# EDDI: Efficient Dynamic Discovery of High-Value Information with Partial VAE

## Abstract

Making decisions requires information relevant to the task at hand. Many real-life decision making situations allow acquiring further relevant information at a specific cost. For example, in assessing the health status of a patient we may decide to take additional measurements such as diagnostic tests or imaging scans before making a final assessment. More information that is relevant allows for better decisions but it may be costly to acquire all of this information. How can we trade off the desire to make good decisions with the option to acquire further information at a cost? To this end, we propose a principled framework, named *EDDI* (Efficient Dynamic Discovery of high-value Information), based on the theory of Bayesian experimental design. In EDDI we propose a novel *partial variational autoencoder* (Partial VAE), to efficiently handle missing data over varying subsets of known information. EDDI combines this Partial VAE with an acquisition function that maximizes expected information gain on a set of target variables. EDDI is efficient and demonstrates that dynamic discovery of high-value information is possible; we show cost reduction at the same decision quality and improved decision quality at the same cost in benchmarks and in two health-care applications. We believe there is great potential for realizing these gains in real-world decision support systems.

## 1 Introduction

Imagine that a person walks into a hospital with a broken arm. The first question from health-care personnel would be: "How did you break the arm?" instead of "Do you have a cold?", because the answer reveals relevant information for this patient. Human experts dynamically acquire information based on the current understanding of the situation. Automating this human expertise of asking relevant questions is difficult. In other applications such as online questionnaires for example, most existing online questionnaire system either present exhaustive questions (Lewenberg et al., 2017; Shim et al., 2018) or use extremely time-consuming human labeling work to manually build a decision tree for a reduced number of questions (Zakim et al., 2008). This wastes the valuable time of experts or users (patients). An automated solution for personalized dynamic acquisition of information has great potential to save much of this time in many real-life applications.

What are the technical challenges to build an intelligent information acquisition system? *Missing data is a key issue*: taking the questionnaire scenario as an example, at any point in time we only observe a small subset of answers yet have to reason about possible answers for the remaining questions. We thus need an accurate probabilistic model that can perform inference given a variable subset of observed answers. *Another key problem is deciding what to ask next*: this requires assessing the worth of each possible question or measurement, the exact computation of which is intractable. However, compared to current active learning methods we select individual features, not instances; therefore, existing methods are not applicable. In addition, these traditional methods are often not scalable to the large volume of data available in many practical cases (Settles, 2012).

We propose the EDDI (Efficient Dynamic Discovery of high-value Information) framework as a scalable information acquisition system for any given task. We assume that information acquisition is always associated with cost. Given a task, such as estimating the costumers' experience or assessing population health status, we dynamically decide which piece of information to acquire next. The framework is very general, and the information can be presented in any form such as answers to questions, or values of a lab test. Our contributions are:

- We propose a novel efficient information acquisition framework, EDDI (Section 3). To enable EDDI, we contribute technically:
  1. *A partial amortized inference method with different specifications for the inference network (Section 3.2).*
     We extend a current amortized inference method, the variational autoencoder (VAE) (Kingma & Welling, 2014; Rezende et al., 2014), to account for partial observations. The resulting method, which we call Partial VAE, is inspired by the set formulation of the data (Qi et al., 2017; Zaheer et al., 2017). Partial VAE, as a probabilistic framework, is highly scalable, and serves as the base for the EDDI framework. However, Partial VAE itself is generic and can be used on its own as a non-linear probabilistic framework for missing-data imputation.
  2. *An information theoretic acquisition function with an efficient approximation, yielding a novel variable-wise active learning method (Section 3.3).*
     Based on the partial VAE, we select the unobserved variable which contributes most to the task, such as health assessment, evaluated using the mutual information. This acquisition function does not have an analytical solution and we derive an efficient approximation.
- We demonstrate the performance of EDDI on various settings, and apply it in real-life health-care scenarios (Section 4).
  1. We first show the superior performance of the Partial VAE framework on an image inpainting task (Section 4.1).
  2. We then use 6 different datasets from the Machine Learning repository of University of Irvine (UCI) (Dheeru & Karra Taniskidou, 2017) to demonstrate the behavior of EDDI, comparing with multiple baseline methods (Section 4.2).
  3. Finally, we evaluate EDDI on two real-life health-care applications: risk assessment in intensive care (Section 4.3) and public health assessment with national survey (Section 4.4), where traditional methods without amortized inference do not scale. EDDI shows clear improvements in these two applications.

## 2 RELATED WORK

EDDI requires a method that handles partially observed data to enable dynamic variable wise active learning. We thus review related methods for handling partial observation and doing active learning.

### 2.1 PARTIAL OBSERVATION

Missing data entries are common in many real-life applications, which has created a long history of research on the topic of dealing with missing data (Rubin, 1976; Dempster et al., 1977). We describe existing methods below:

**Traditional methods without amortization** Prediction based methods have shown advantages for missing value imputation (Scheffer, 2002). Efficient matrix factorization based methods have been recently applied (Keshavan et al., 2010; Jain et al., 2010; Salakhutdinov & Mnih, 2008), where the observations are assumed to be able to decompose as multiplication of low dimensional matrices. In particular, many probabilistic frameworks with various distribution assumptions (Salakhutdinov & Mnih, 2008; Blei et al., 2003) have been used for missing value imputation (Yu et al., 2016; Hamesse et al., 2018) and also recommender systems where unlabeled items are predicted (Stern et al., 2009; Wang & Blei, 2011; Gopalan et al., 2014).

The probabilistic matrix factorization method has been used in the active variable selection framework, the dimensionality reduction active learning model (DRAL),(Lewenberg et al., 2017). These traditional methods suffer from limited model capacity since they are commonly linear. Additionally, they do not scale to large volumes of data and thus are usually not applicable in real-world applications. For example, Lewenberg et al. (2017) test the performance of their method with a single user due to the heavy computational cost of traditional inference methods for probabilistic matrix factorization.

**Utilizing Amortized Inference** The amortized inference (Kingma & Welling, 2014; Rezende et al., 2014; Zhang et al., 2017) has significantly improved the scalability for probabilistic models such as variational autoencoders (VAEs). In the case of partially observed data, amortized inference is particularly of interest due to the need of speeding up test time applications. Wu et al. (2018) employ traditional non-amortized inference in order to perform partial inference of a pretrained VAE during

test time. Amortized inference is only used during training, assuming the training dataset is fully observed. During test time, the traditional inference is used to infer missing data entries from the partially observed dataset using the pre-trained model. In this way, only training time is reduced. The model is restrictive since it is not scalable in the test time and the fully observed training set is not available for many applications.

Nazabal et al. (2018) uses zero imputation (ZI) for amortized inference for both training and test sets with missing data entries. ZI is a generic and straightforward method that first fills the missing data with zeros, and then feeds the imputed data as input for the inference network. The drawback of zero imputation is that it introduces bias when the data are not missing completely at random which leads to not well-learned model. We also observe artifacts when using it for the image inpainting task. In the end, independent of our work, Garnelo et al. (2018) explore interpreting variational autoencoder (amortized inference) as stochastic processes, which also handles partial observation per se.

## 2.2 ACTIVE LEARNING

**Traditional Active Learning** Active learning, also referred to as experimental design, aims to obtain optimal performance with fewer selected data (or experiments) (Lindley, 1956; MacKay, 1992; Settles, 2012). Traditional active learning aims to select the next data point to label. Many information theoretical approaches have shown promising results in various settings with different acquisition functions (MacKay, 1992; McCallumzy & Nigamy, 1998; Houlsby et al., 2011). These methods commonly assume that there exist fully observed data, and the acquisition decision is instance wise. Little work has dealt with missing values within instances. Zheng & Padmanabhan (2002) deal with missing data values by imputing with traditional non-probabilistic methods (Little & Rubin, 1987) first. It is still an instance-wise active learning framework.

Different from traditional active learning, our proposed framework aims to for perform variable-wise active learning for each instance. In this setting, information theoretical acquisition functions need a new design as well as non-trivial approximations. The most closely related work is the aforementioned DRAL (Lewenberg et al., 2017), which deals with variable-wise active learning for each instance.

**Active Feartue Acquisition (AFA)** Active sequential feature selection is of great need, especially in cost-sensitive applications. Thus, many methods have also been applied and resulted in the class of methodologies called Active Feature Acquisition (AFA) (Melville et al., 2004; Saar-Tsechansky et al., 2009; Thahir et al., 2012; Huang et al., 2018). For instance, Melville et al. (2004); Saar-Tsechansky et al. (2009) have designed objectives to select any feature from any instance to minimize the cost to archive high accuracy. The proposed framework is very general. However, the problem setting of AFA methods are entirely different from our active variable selection problem: AFA mainly studies the optimization of *optimal training set* that would result in the best classifier (model), under limited budget of costs, while our framework studies a slightly different problem: given a pretrained model, how to identify and acquire high value information with minimal costs. Hence, AFA can not be directly applied. Also, AFA requires fully observed variables at test time, while our framework does not require this assumption. Last but not the least, the realization of these framework relies on various heuristics and suffer from limited scalability.

## 3 METHOD

In this section, we first formalize the active variable selection problem that we aim to solve. Then, we present our Partial VAE to model and perform inference on partial observations. Finally, we complete the EDDI framework by presenting our acquisition function and estimation method.

### 3.1 PROBLEM FORMULATION

The core problem that we address in this paper is the following active variable selection problem. Let $\mathbf{x} = [x_1, \ldots, x_{|I|}]$ be a set of random variables with probability density $p(\mathbf{x})$. Furthermore, let a subset of the variables $\mathbf{x}_O$, $O \subset I$, be observed while the variables $\mathbf{x}_U$, $U = I \setminus O$, are unobserved. We assume that we can query the value of variables $x_i$ for $i \in U$. The goal of active variable selection is to query a sequence of variables in $U$ with the goal of predicting a quantity of interest $f(\mathbf{x})$, as accurately as possible while simultaneously performing as little queries as possible, where $f(\cdot)$ can be any (random) function. This problem, in the simplified myopic setting, can be formalized as that

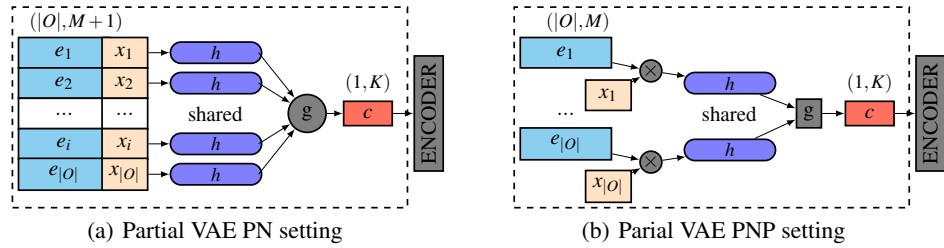

(a) Partial VAE PN setting          (b) Parial VAE PNP setting

Figure 1: Illustration of Partial VAE encoder architecture.

of proposing the next variable $x_{i^*}$ to be queried by maximizing a reward function $R$, i.e.

$$i^* = \arg\max_{i \in U} R(i \mid \mathbf{x}_O) \tag{1}$$

where $R(i \mid \mathbf{x}_O)$ quantifies the merit of our prediction of $f(\cdot)$ given $\mathbf{x}_0$ and $x_i$. Furthermore, the reward can quantify other properties important to the problem, e.g. the cost of acquiring $x_i$.

## 3.2 PARTIAL AMORTIZATION OF INFERENCE QUERIES

We first introduce how to establish a generative probabilistic model of random variables $\mathbf{x}$, that is capable of handling unobserved (missing) variables $\mathbf{x}_U$ with variable size. Our approach to this, named the Partial VAE, is based on the Variational autoencoder (VAE), which enables inference to scale to large volumes of data.

**VAE and amortized inference** VAE defines a generative model where the data $\mathbf{x}$ are generated from latent variables $\mathbf{z}$, defined as $p(\mathbf{x}, \mathbf{z}; \theta) = \prod_n p(\mathbf{x}_n|\mathbf{z}_n; \theta) p(\mathbf{z}_n)$. The data generation, $p_\theta(\mathbf{x}_n|\mathbf{z}_n)$, is realized by a deep neural network. To approximate the the posterior of the latent variable $p_\theta(\mathbf{z}_n|\mathbf{x}_n)$, VAE uses *amortized* variational inference. Specifically, it uses an encoder, which is another neural network with the data $\mathbf{x}_n$ as input to produce the variational approximation of the posterior $q(\mathbf{z}_n|\mathbf{x}_n; \phi)$. As traditional variational inference, VAE is trained by maximizing an evidence lower bound (ELBO), which is equivalent to minimize the KL divergence between $p_\theta(\mathbf{z}_n|\mathbf{x}_n)$ and $q(\mathbf{z}_n|\mathbf{x}_n; \phi)$.

VAE is not directly applicable to data with missing values. Consider a partitioning that divides the variables into *observed* variables $\mathbf{x}_O$ and *unobserved* variables $\mathbf{x}_U$. In this setting, we would like to efficiently and accurately infer $p(\mathbf{z}|\mathbf{x}_O)$ and $p(\mathbf{x}_U|\mathbf{x}_O)$. One challenge in the above setting is that there are many possible partitioning of $\{U, O\}$, where the size of observed ratings might vary. Therefore, classic approaches to train a VAE with variational bound and *amortize* inference networks are no longer directly applicable. We propose to extend amortization to our *partial* inference situation.

**Partial VAE** In a VAE, a factorized structure for $p(\mathbf{x}|\mathbf{z})$ is always assumed, i.e.

$$p(\mathbf{x}|\mathbf{z}) = \prod_i p_i(\mathbf{x}_i|\mathbf{z}). \tag{2}$$

This implies that given $\mathbf{z}$, the observed variables $\mathbf{x}_O$ are conditionally independent of $\mathbf{x}_U$. Therefore,

$$p(\mathbf{x}_U|\mathbf{x}_O, \mathbf{z}) = p(\mathbf{x}_U|\mathbf{z}), \tag{3}$$

and inferences about $\mathbf{x}_U$ can be reduced to inference about $\mathbf{z}$. Therefore, the key object of interest in this setting is $p(\mathbf{z}|\mathbf{x}_O)$, i.e., the posterior over the shared latent variables $\mathbf{z}$ given the observed variables $\mathbf{x}_O$. Once knowledge about $\mathbf{z}$ is obtained, we can draw correct inferences about $\mathbf{x}_U$. To approximate $p(\mathbf{z}|\mathbf{x}_O)$ we introduce an auxiliary variational inference network $q(\mathbf{z}|\mathbf{x}_O)$ and define a partial variational upper bound,

$$
\begin{aligned}
D_{\mathrm{KL}}(q(\mathbf{z}|\mathbf{x}_O)\|p(\mathbf{z}|\mathbf{x}_O)) &= \mathbb{E}_{\mathbf{z}\sim q(\mathbf{z}|\mathbf{x}_O)}[\log q(\mathbf{z}|\mathbf{x}_O) - \log p(\mathbf{z}|\mathbf{x}_O)] \\
&\leq \mathbb{E}_{\mathbf{z}\sim q(\mathbf{z}|\mathbf{x}_O)}[\log q(\mathbf{z}|\mathbf{x}_O) - \log p(\mathbf{x}_O|\mathbf{z}) - \log p(\mathbf{z})] \equiv \mathscr{L}_{partial}.
\end{aligned} \tag{4}
$$

This bound, $\mathscr{L}_{partial}$, depends only on the observation $\mathbf{x}_O$, which could vary between different data points. We call the auxiliary distribution $q(\mathbf{z}|\mathbf{x}_O)$ the *partial inference net* since it takes a set of partially observed variables $\mathbf{x}_O$ whose length may vary. Specifying $q(\mathbf{z}|\mathbf{x}_O)$ requires distribution over random partitioning $\{O, U\}$.

**Amortized Inference with partial observations** Inspired by the *Point Net (PN)* approach for point cloud classification (Qi et al., 2017; Zaheer et al., 2017), we specify the approximate distribution $q(\mathbf{z}|\mathbf{x}_O)$ by a *permutation invariant set function encoding*, given by:

$$\mathbf{c}(\mathbf{x}_O) := g(h(\mathbf{s}_1), h(\mathbf{s}_2), ..., h(\mathbf{s}_{|O|})), \tag{5}$$

where $|O|$ is the number of the observed variables, $\mathbf{s}_d$ carries the information of the input identify $\mathbf{e}_d$ and the input value $x_d$. There are many ways to define $\mathbf{e}_d$. Naively, it could be the coordinates for points in the point cloud, and one-hot embedding of the number of questions in a questionnaire. With different problem settings, it can be beneficial to learn $\mathbf{e}$ as an embedding of the identity of the variable, either with or without an naive encoding as input. In this work, we treat $\mathbf{e}$ as an unknown embedding, which is optimized during training process.

There are also different ways to construct $\mathbf{s}_d$. Concatenation, $\mathbf{s}_d = [\mathbf{e}_d, x_d]$, is commonly used in computer vision applications (Qi et al., 2017). Such architecture is illustrated in Figure 1(a). However, we note that the construction of $\mathbf{s}_d$ can be flexible. We propose to construct $\mathbf{s} = \mathbf{e}_d * x_d$ using element-wise multiplication, shown in Figure 1(b). We show that this formulation generalizes naive Zero Imputation (ZI) VAE (Nazabal et al., 2018). We call this approach *Pointnet Plus (PNP)* specification of Partial VAE. The theoretical consideration of relating ZI to PNP is presented in Appendix C.1.

We then use a neural network $h(\cdot)$ to map input $\mathbf{s}$ from $\mathbb{R}^{M+1}$ to $\mathbb{R}^K$, where $M$ is the dimension of each $\mathbf{e}_d$, $x_d$ is a scalar, and $K$ is the latent space size. Key to the PN structure is the permutation invariant aggregation operation $g(\cdot)$, such as max-pooling or summation. In this way, the mapping $\mathbf{c}(\mathbf{x}_O)$ is invariant to permutations of elements of $\mathbf{x}_O$ and $\mathbf{x}_O$ can have arbitrary length. Finally, the fixed-size code $\mathbf{c}(\mathbf{x}_O)$ is fed into an ordinary amortized inference net, that transforms the code into the statistics of a multivariate Gaussian distribution to approximate $p(\mathbf{z}|\mathbf{x}_O)$. The procedure is illustrated in the first dashed box in Figure 1, which is our basic Partial VAE method.

### 3.3 Efficient Dynamic Discovery of High-value Information

We now cast the active variable selection problem (1) as adaptive Bayesian experimental design, utilizing $p(\mathbf{x}_U|\mathbf{x}_O)$ inferred by Partial VAE. Algorithm 1 summarize the EDDI framework.

**Information Reward** We designed a variable selection acquisition function in an information theoretical way following Bayesian experimental design (Lindley, 1956; Bernardo, 1979). Lindley (1956) provides a generic formulation of Bayesian experimental design by maximizing the expected Shannon information. Bernardo (1979) generalizes it by considering the decision task context.

For a given task, we may be interested in statistics of some variables $\mathbf{x}_\phi$, where $\mathbf{x}_\phi \subset \mathbf{x}_U$. Given a new instance (user), assume we have observed $\mathbf{x}_O$ so far for this instance, and we need to select the next variable $x_i$ (an element of $\mathbf{x}_{U \setminus \phi}$) to observe. Following Bernardo (1979), We select $x_i$ by maximizing:

$$R(i, \mathbf{x}_O) = \mathbb{E}_{\mathbf{x}_i \sim p(\mathbf{x}_i|\mathbf{x}_O)} D_{\mathrm{KL}} \left[ p(\mathbf{x}_\phi|\mathbf{x}_i, \mathbf{x}_O) \,\|\, p(\mathbf{x}_\phi|\mathbf{x}_O) \right]. \tag{6}$$

In our paper, we mainly consider the case that a subset of interesting observations represents the statistics of interest $\mathbf{x}_\phi$. Sampling $\mathbf{x}_i \sim p(\mathbf{x}_i|\mathbf{x}_o)$ is approximated by $\mathbf{x}_i \sim \hat{p}(\mathbf{x}_i|\mathbf{x}_o)$, where $\hat{p}(\mathbf{x}_i|\mathbf{x}_o)$ is defined by the following process in Partial VAE. It is implemented by first sampling $\mathbf{z} \sim q(\mathbf{z}|\mathbf{x}_o)$, and then $\mathbf{x}_i \sim p(\mathbf{x}_i|\mathbf{z})$. The same applies for $p(\mathbf{x}_i, \mathbf{x}_\phi|\mathbf{z})$ appeared in Equation 9.

**Efficient approximation of the Information reward** The Partial VAE allows us to sample $\mathbf{x}_i \sim p(\mathbf{x}_i|\mathbf{x}_o)$. However, the KL term in Equation 6,

$$D_{KL} \left[ p(\mathbf{x}_\phi|\mathbf{x}_i, \mathbf{x}_o) || p(\mathbf{x}_\phi|\mathbf{x}_o) \right] = - \int_{\mathbf{x}_\phi} p(\mathbf{x}_\phi|\mathbf{x}_i, \mathbf{x}_o) \log \frac{p(\mathbf{x}_\phi|\mathbf{x}_o)}{p(\mathbf{x}_\phi|\mathbf{x}_i, \mathbf{x}_o)}, \tag{7}$$

is intractable to evaluate since both $p(\mathbf{x}_\phi|\mathbf{x}_i, \mathbf{x}_o)$ and $p(\mathbf{x}_\phi|\mathbf{x}_o)$ are intractable. For high dimensional $\mathbf{x}_\phi$, entropy estimation could be difficult. The entropy term $\int_{\mathbf{x}_\phi} p(\mathbf{x}_\phi|\mathbf{x}_i, \mathbf{x}_o) \log p(\mathbf{x}_\phi|\mathbf{x}_i, \mathbf{x}_o)$ depends on $i$ hence cannot be ignored. In the following, we show how to approximate this expression.

Our proposal is based on the observation that analytic solutions of KL-divergences are available under specific variational distribution families of $q(\mathbf{z}|\mathbf{x}_O)$ (such as the Gaussian distribution commonly used in VAEs). Instead of calculating information reward in $\mathbf{x}$ space, we have shown that one can equivalently perform calculations in $\mathbf{z}$ space (cf. Appendix A.1):

$$R(i, \mathbf{x}_o) = \mathbb{E}_{\mathbf{x}_i \sim p(\mathbf{x}_i|\mathbf{x}_o)} D_{KL} [p(\mathbf{z}|\mathbf{x}_i, \mathbf{x}_o) || p(\mathbf{z}|\mathbf{x}_o)] \tag{8}$$

$$- \mathbb{E}_{\mathbf{x}_\phi, \mathbf{x}_i \sim p(\mathbf{x}_\phi, \mathbf{x}_i|\mathbf{x}_o)} D_{KL} \left[ p(\mathbf{z}|\mathbf{x}_\phi, \mathbf{x}_i, \mathbf{x}_o) || p(\mathbf{z}|\mathbf{x}_\phi, \mathbf{x}_o) \right].$$

---

**Algorithm 1** EDDI: Algorithm Overview

---

**Require:** Training dataset $\mathbf{x}_{trn}$, which is partially observed; Test dataset $\mathbf{x}_{tst}$ without any observation; Indices $\phi$ of target variables.
 1: **Train Partial VAE** by optimizing partial variational bound with $\mathbf{x}_{trn}$ (cf. Section 3.2)
 2: **Actively acquire feature value** $x_i$ to estimate $\mathbf{x}_\phi$ for each test point (cf. Section 3.3)
  **for** each test instance **do**
    $\mathbf{x}_O \leftarrow \emptyset$ (no variable value has been observed for any test point)
    **repeat**
      Choose variable $x_i$ from $U \setminus \phi$ to maximize the information reward (Equation 9)
      $\mathbf{x}_O \leftarrow x_i \cup \mathbf{x}_O$
    **until** Stopping criterion reached (e.g. the time budget)
  **end for**

---

Table 1: Comparing models trained on partially observed MNIST. VAE-full is an ideal reference.

| Method | VAE-full | ZI | ZI-m | PN | PNP |
|---|---|---|---|---|---|
| Train ELBO | -95.05 | **-113.64** | -117.29 | -121.43 | **-113.64** |
| Test ELBO (Rnd.) | -101.46 | -116.01 | -118.61 | -122.20 | **-114.01** |
| Test ELBO (Reg.) | -101.46 | -130.61 | -123.87 | -116.53 | **-113.19** |

Note that Equation 8 is exact. Additionally, we use partial VAE approximation $p(\mathbf{z}|\mathbf{x}_\phi,\mathbf{x}_i,\mathbf{x}_o) \approx q(\mathbf{z}|\mathbf{x}_\phi,\mathbf{x}_i,\mathbf{x}_o)$, $p(\mathbf{z}|\mathbf{x}_o) \approx q(\mathbf{z}_i|\mathbf{x}_o)$ and $p(\mathbf{z}|\mathbf{x}_i,\mathbf{x}_o) \approx q(\mathbf{z}_i|\mathbf{x}_i,\mathbf{x}_o)$. This leads to the final approximation of the information reward:

$$\hat{R}(i,\mathbf{x}_o) = \mathbb{E}_{\mathbf{x}_i \sim \hat{p}(\mathbf{x}_i|\mathbf{x}_o)} D_{KL}\left[q(\mathbf{z}|\mathbf{x}_i,\mathbf{x}_o)||q(\mathbf{z}|\mathbf{x}_o)\right] \tag{9}$$
$$- \mathbb{E}_{\mathbf{x}_\phi,\mathbf{x}_i \sim \hat{p}(\mathbf{x}_\phi,\mathbf{x}_i|\mathbf{x}_o)} D_{KL}\left[q(\mathbf{z}|\mathbf{x}_\phi,\mathbf{x}_i,\mathbf{x}_o)||q(\mathbf{z}|\mathbf{x}_\phi,\mathbf{x}_o)\right].$$

With this approximation, the divergence between $q(\mathbf{z}|\mathbf{x}_i,\mathbf{x}_o)$ and $q(\mathbf{z}|\mathbf{x}_o)$ can often computed analytically in Partial VAE setting, for example, under Gaussian parameterization. The only Monte Carlo sampling required is the one set of samples $\mathbf{x}_\phi,\mathbf{x}_i \sim p(\mathbf{x}_\phi,\mathbf{x}_i|\mathbf{x}_o)$ that can be shared across different KL terms in Equation 9.

## 4 EXPERIMENTS

We evaluate our proposed EDDI framework with various settings. We first assess the Partial VAE component of EDDI alone on an image inpainting task both qualitatively and quantitatively (Section 4.1). We compare our proposed two PN-based Partial VAE with the zero-imputing (ZI) VAE (Nazabal et al., 2018). Additionally, we modify ZI VAE to use s mask matrix indicating which variables are currently observed as input. We name this method ZI-m VAE. We then demonstrate the performance of the entire EDDI framework on datasets from the UCI repository (Section 4.2 ), as well as in two real-life application scenarios: Risk assessment in intensive care (Section 4.3) and public health assessment with national health survey (Section 4.4). We compare the performance of EDDI, using four different Partial VAE settings, with three baselines. The first baseline is the *random active feature selection strategy (denoted as RAND)* which randomly picks the next variable to observe. The second baseline method is the *single best strategy (denoted as SING)* which finds a single global optimal order of picking up variables. This order is then applied to all data points. SING uses the objective function as in Equation (9) to find the optimal ordering by averaging over all the data.

### 4.1 IMAGE INPAINTING WITH PARTIAL VAE

We evaluate the performance of Partial VAE with the image inpainting task, which is to fill in the removed pixels in an image. We perform the evaluation in two different settings: We remove the pixels at random in the first setting, and remove a region of the pixels in the second setting.

**Inpainting Random Missing Pixels** We use MNIST dataset (LeCun, 1998) and remove pixels randomly for this task. The same setting are used for all methods (see Appendix B.1 for details). During training, we remove a random portion (uniformly sampled between 0% and 70%) of pixels. We then impute missing pixels on a partially observed test set (constructed by removing 70% of the pixels). The performance of pixel imputation is evaluated by test ELBOs on missing pixels. The first two rows in Table 1 show training and test ELBOs for all algorithms using this partially observed

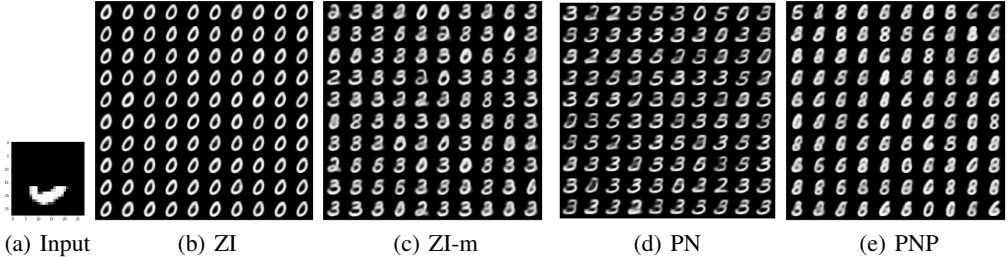

| (a) Input | (b) ZI | (c) ZI-m | (d) PN | (e) PNP |

Figure 2: Image inpainting example with MNIST dataset using Partial VAE with four settings.

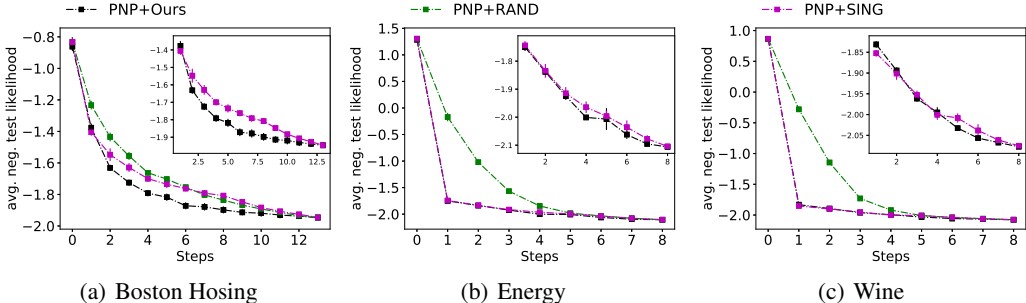

| (a) Boston Hosing | (b) Energy | (c) Wine |

Figure 3: Information curves of active variable selection, demonstrated on three UCI datasets (based on PNP parameterization of Partial VAE). This displays negative test log likelihood (y axis, the lower the better) during the course of active selection (x-axis). Error bars represent standard errors.

dataset. Additionally, we show ordinary VAE (VAE-full) trained on the fully observed dataset as an ideal reference. Among all Partial VAE methods, the PNP approach performs best.

**Inpainting Regions** We then consider inpainting large contiguous regions of images. It aims to evaluate the capability of the Partial VAEs to produce all possible outcomes with better uncertainty estimates. With the same trained model as before, we remove the region of the upper 60% pixels of the image in the test set. We then evaluate the average likelihoods of the models. The last row of Table 1 shows the results of the test ELBO in this case. PNP based Partial VAE performs better than other settings. Note that given only the lower half of a digit, the number cannot be identified uniquely. ZI (Figure 2(b)) fails to cover the different possible modes due to its limitation in posterior inference. ZI-m (Figure 2(c)) is capable of producing multiple modes. However, some of the generated samples are not consistent with the given part (i.e., some digits of 2 are generated). Our proposed PN (Figure 2(d)) and PNP Figure 2(e)) are capable of recovering different modes, and are consistent with observations.

## 4.2 EDDI ON UCI DATASETS

Given the effectiveness of our proposed Partial VAE, we now demonstrate the performance of our proposed EDDI framework in comparison with random selection (RAND) and single optimal ordering (SING). We first apply EDDI on 6 different UCI datasets (cf. Appendix B.2) (Dheeru & Karra Taniskidou, 2017). We report the results of EDDI with all these four different specifications of Partial VAE (ZI, ZI-m, PN, PNP).

All Partial VAE are first trained on partially observed UCI datasets where a random portion of variables is removed. We actively select variable for each test point starting with empty observation $\mathbf{x}_o = \emptyset$. In all UCI datasets, We randomly sample 10% of the data as the test set. All experiments repeated for ten times.

Taking PNP based setting as an example, Figure 3 shows the negative test log likelihood on $\mathbf{x}_\phi$ for each variable selection step with three different datasets, where $\mathbf{x}_\phi$ is defined by the UCI task. We call this curve the *information curve (IC)*. We see that EDDI can obtain information efficiently. It archives the same negative test log likelihood with less than half of the variables. Single optimal ordering also improves upon random ordering. However, it is less efficient compared with EDDI since EDDI perform active learning for each data instance which is "personalized". Figure 4 shows

Table 2: Average ranking of AUIC over 6 UCI datasets.

| Method | ZI | ZI-m | PNP | PN |
|---|---|---|---|---|
| EDDI | 5.72 (0.03 ) | 5.54 (0.02 ) | **5.08 (0.02 )** | 5.25 (0.02) |
| Random | 8.03 (0.03 ) | 8.10 (0.03 ) | 7.77 (0.03 ) | 7.79 (0.03 ) |
| Single best | 8.68 (0.03 ) | 5.50 (0.02 ) | 5.20 (0.02 ) | 5.28 (0.02 ) |

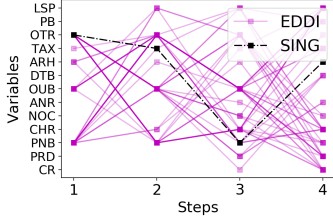

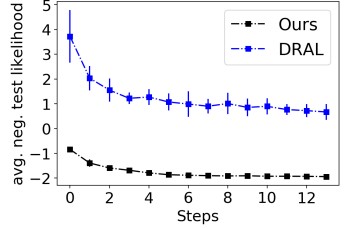

| Method | Time |
|---|---|
| DRAL | 2747.16 |
| EDDI | **2.64** |

Figure 4: First four decision steps on Boston Housing test data. EDDI is "personalized" comparing SING. Full names of the variables are listed in the Appendix B.2.

Figure 5: Comparison of DRAL (Lewenberg et al., 2017) and EDDI on Boston Housing dataset. EDDI out performs DRAL significantly regarding test log likelihood in every step.

Table 3: Test CPU time (in seconds) per test point for active variable selection using EDDI and DRAL. EDDI is $10^3$ times more computation efficient than DRAL (Lewenberg et al., 2017).

an example of the decision processes using EDDI and SING. The first step of EDDI overlaps largely with SING. From the second step, EDDI makes "personalized" decisions.

We also present the average performance among all datasets with different settings. The area under the information curve (AUIC), $\sum_t -\log p(\mathbf{x}_\phi|\mathbf{x}_{O_t})$, can then be used to compare the performance across models and strategies. Smaller AUIC value (could be positive or negative) indicates better performance. However, due to different datasets have different scales of test likelihoods and different numbers of variables (indicated by steps), it is not fair to average the AUIC across datasets to compare overall performances. We thus define average *ranking* of AUIC that compares 12 methods (indexed by $i$) averaging these datasets as: $r_i = \frac{1}{\sum_j N_j} \sum_{j=1}^6 \sum_{k=1}^{N_j} r_{ijk}, \ i = 1, .., 16$. These 12 methods are cross combinations of four Partial VAE models with three variable selection strategies. $r_i$ is the final ranking of $i$th combination, $r_{ijk}$ is the ranking of the $i$th combination (based on AUIC value) regarding the $k$th test data point in the $j$th UCI dataset, and $N_j$ the size of the $j$th UCI dataset. This gives us $6\sum_j N_j$ different rankings. Finally, we simply compute the mean and standard error statistics based on these rankings. Table 2 summarize the average ranking results. We provide additional statistical significance test (Wilcoxcon signed-rank test for paired data) in Appendix B.2.3. We can conclude that EDDI outperforms other variable selection order in all different Partial VAE settings. Among different partial VAE settings, PNP/PN-based settings perform better than ZI-based settings.

**Comparison with non-amortized method** Additionally, we compare EDDI to DRAL (Lewenberg et al., 2017) which is the state-of-the-art method for the same problem setting. As discussed in Section 2, DRAL is linear and requires high computational cost. The DRAL paper only tested their method on a single test data point due to its limitation on computational efficiency. We compare DRAL with EDDI on Boston Housing dataset with ten randomly selected test points here. Results are shown in Figure 5, where EDDI significantly outperforms DARL thanks to more flexible Partial VAE model. Additionally, EDDI is 1000 times more efficient than DARL as shown in Table 3.

### 4.3 RISK ASSESSMENT WITH MIMIC-III

We now apply EDDI to risk assessment tasks using the Medical Information Mart for Intensive Care (MIMIC III) database (Johnson et al. (2016)). MIMIC III is the most extensive publicly available clinical database, containing real-world records from over 40,000 critical care patients with 60,000 ICU stays. The risk assessment task is to predict the final mortality. We preprocess the data for this task following Harutyunyan et al. (2017) [1]. This results in a dataset of 21139 patients. We treat the final mortality of a patient as a Bernoulli variable. For our task, we focus on variable selection, which corresponds to medical instrument selection. We thus further process the time series variables into static variables based on temporal averaging.

---

[1] https://github.com/yerevann/mimic3-benchmarks

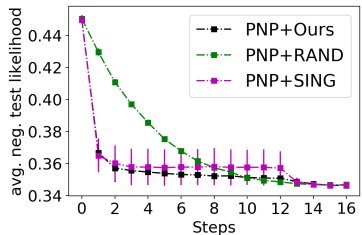

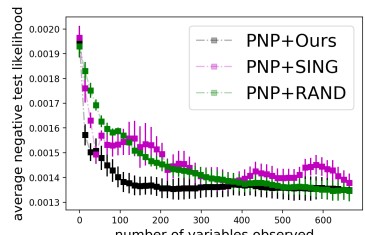

Figure 6: Information curves of active variable selection on risk assessment task on MIMIC III, produced with PNP setting.

Figure 7: Information curves of active (grouped) variable selection on risk assessment task on NHANES, produced with PNP setting.

| Method | EDDI | Random | Single best |
|--------|------|--------|-------------|
| ZI | 5.28 (0.01) | 7.12 (0.02) | 6.28 (0.01) |
| ZI-m | 5.82 (0.01) | 7.95 (0.01) | 6.82 (0.01) |
| PN | 5.24 (0.01) | 7.91 (0.01) | 6.24 (0.01) |
| PNP | **5.23 (0.01)** | 7.82 (0.01) | 6.23 (0.01) |

| Method | EDDI | Random | Single best |
|--------|------|--------|-------------|
| ZI | 5.68 (0.13) | 8.44 (0.13) | 6.36 (0.13) |
| ZI-m | 7.63 (0.12) | 8.69 (0.12) | 8.97 (0.12) |
| PN | 5.64 (0.16) | 6.14 (0.15) | 5.56 (0.16) |
| PNP | **4.41 (0.12)** | 5.34 (0.14) | 5.13 (0.12) |

Table 4: Average ranking on AUIC of MIMIC III

Table 5: Average ranking on AUIC of NHANES

Figure 6 shows the information curve of different strategies, using PNP based Partial VAE as an example (more results in Appendix B.3). Table 4 shows the average ranking of AUIC with different settings. In this application, EDDI significantly outperforms other variable selection strategies in all different settings of Partial VAE, and PNP based setting performs best.

## 4.4 PUBLIC HEALTH ASSESSMENT WITH NHANES

Finally, we apply our methods to public health assessment using NHANES 2015-2016 data cdc (2005). NHANES is a program with adaptable components of measurements, to assess the health and nutritional status of adults and children in the United States. Every year, approximately thousands individuals of all ages are interviewed in their homes and complete the health examination component of the survey. This 2015-2016 NHANES data contains three major sections, the questionnaire interview, examinations and lab tests for 9971 subjects in the publicly available version of this cycle. In our setting, we consider the whole set of lab test results (139 dimensions of variables) as the target variable of interest $\mathbf{x}_\phi$ since they are expensive and reflects the subject's health status, and we active select the questions from the extensive questionnaire (665 variables).

In NHANES, the entire questionnaire is divided into 73 different groups. In practice, questions in the same group are often examined together. Therefore, we perform active variable selection on the group level: at each step, the algorithm will be selecting one group to observe. This is more challenging than the experiments in previous sections since it requires the generative model to simulate a group of unobserved data in Equation (9) at the same time. When evaluating test likelihood on the target variable of interest, we treat variables in each group equally. For a fair comparison, the calculation of the area under the information curve (AUIC) is weighted by the size of the group chosen by the algorithms. Specifically, AUIC is calculated after spline interpolation. The information curve plots in Figure 7, together with Table 5 of AUIC statistics show that our EDDI outperforms other baselines. This experiment shows that EDDI is capable of performing active selection on a large pool of grouped variables to estimate a high dimensional target.

## 5 CONCLUSION

In this paper, we present EDDI, a novel and efficient framework for dynamic active variable selection for each instance. Within the EDDI framework, we propose Partial VAE which performs amortized inference to handle missing data. Partial VAE alone can be used as a non-linear computational efficient probabilistic imputation method. Based on Partial VAE, we design a variable wise acquisition function for EDDI and derived corresponding approximation method. EDDI has demonstrated its effectiveness on active variable selection tasks across multiple real-world applications. In the future, we would extend the EDDI framework to handle more complicated scenarios, such as time-series, or the cold-start situation.

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

# A    ADDITIONAL DERIVATIONS

## A.1    INFORMATION REWARD APPROXIMATION

In our paper, given the VAE model $p(\mathbf{x}|z)$ and a partial inference network $q(\mathbf{z}|\mathbf{x}_o)$, the experimental design problem is formulated as maximization of the information reward:

$$R(i, \mathbf{x}_o) = \mathbb{E}_{\mathbf{x}_i \sim p(\mathbf{x}_i|\mathbf{x}_o)}[D_{KL}(p(\mathbf{x}_\phi|\mathbf{x}_i, \mathbf{x}_o)||p(\mathbf{x}_\phi|\mathbf{x}_o))]$$

Where $p(\mathbf{x}_\phi|\mathbf{x}_i, \mathbf{x}_o) = \int_\mathbf{z} p(\mathbf{x}_\phi|\mathbf{z})q(\mathbf{z}|\mathbf{x}_i, \mathbf{x}_o)$, $p(\mathbf{x}_\phi|\mathbf{x}_o) = \int_\mathbf{z} p(\mathbf{x}_\phi|\mathbf{z})q(\mathbf{z}|\mathbf{x}_o)$ and $q(\mathbf{z}|\mathbf{x}_o)$ are approximate condition distributions given by partial VAE models. Now we consider the problem of directly approximating $R(i, \mathbf{x}_o)$.

Applying the chain rule of KL-divergence, we have:

$$\begin{aligned} D_{KL}(p(\mathbf{x}_\phi|\mathbf{x}_i, \mathbf{x}_o)||p(\mathbf{x}_\phi|\mathbf{x}_o)) = {} & D_{KL}(p(\mathbf{x}_\phi, \mathbf{z}|\mathbf{x}_i, \mathbf{x}_o)||p(\mathbf{x}_\phi, \mathbf{z}|\mathbf{x}_o)) \\ & - \mathbb{E}_{\mathbf{x}_\phi \sim p(\mathbf{x}_\phi|\mathbf{x}_i, \mathbf{x}_o)}\left[ D_{KL}(p(\mathbf{z}|\mathbf{x}_\phi, \mathbf{x}_i, \mathbf{x}_o)||p(\mathbf{z}|\mathbf{x}_\phi, \mathbf{x}_o)) \right], \end{aligned}$$

Using again the KL-divergence chain rule on $D_{KL}(p(\mathbf{x}_\phi, \mathbf{z}|\mathbf{x}_i, \mathbf{x}_o)||p(\mathbf{x}_\phi, \mathbf{z}|\mathbf{x}_o))$, we have:

$$\begin{aligned} & D_{KL}(p(\mathbf{x}_\phi, \mathbf{z}|\mathbf{x}_i, \mathbf{x}_o)||p(\mathbf{x}_\phi, \mathbf{z}|\mathbf{x}_o)) \\ = {} & D_{KL}(p(\mathbf{z}|\mathbf{x}_i, \mathbf{x}_o)||p(\mathbf{z}|\mathbf{x}_o)) + D_{KL}(p(\mathbf{x}_\phi|\mathbf{z}, \mathbf{x}_i, \mathbf{x}_o)||p(\mathbf{x}_\phi|\mathbf{z}, \mathbf{x}_o)) \\ = {} & D_{KL}(p(\mathbf{z}|\mathbf{x}_i, \mathbf{x}_o)||p(\mathbf{z}|\mathbf{x}_o)) + D_{KL}(p(\mathbf{x}_\phi|\mathbf{z})||p(\mathbf{x}_\phi|\mathbf{z})) \\ = {} & D_{KL}(p(\mathbf{z}|\mathbf{x}_i, \mathbf{x}_o)||p(\mathbf{z}|\mathbf{x}_o)). \end{aligned}$$

The KL-divergence term in the reward formula is now rewritten as follows,

$$\begin{aligned} D_{KL}(p(\mathbf{x}_\phi|\mathbf{x}_i, \mathbf{x}_o)||p(\mathbf{x}_\phi|\mathbf{x}_o)) = {} & D_{KL}(p(\mathbf{z}|\mathbf{x}_i, \mathbf{x}_o)||p(\mathbf{z}|\mathbf{x}_o)) \\ & - \mathbb{E}_{\mathbf{x}_\phi \sim p(\mathbf{x}_\phi|\mathbf{x}_i, \mathbf{x}_o)}\left[ D_{KL}(p(\mathbf{z}|\mathbf{x}_\phi, \mathbf{x}_i, \mathbf{x}_o)||p(\mathbf{z}|\mathbf{x}_\phi, \mathbf{x}_o)) \right]. \end{aligned}$$

One can then plug in the partial VAE inference approximation:

$$p(\mathbf{z}|\mathbf{x}_\phi, \mathbf{x}_i, \mathbf{x}_o) \approx q(\mathbf{z}|\mathbf{x}_\phi, \mathbf{x}_i, \mathbf{x}_o), \ \ p(\mathbf{z}|\mathbf{x}_i, \mathbf{x}_o) \approx q(\mathbf{z}|\mathbf{x}_i, \mathbf{x}_o), \ \ p(\mathbf{z}|\mathbf{x}_o) \approx q(\mathbf{z}|\mathbf{x}_o)$$

Finally, the information reward is now approximated as:

$$\begin{aligned} R(i, \mathbf{x}_o) \approx {} & \mathbb{E}_{\mathbf{x}_i \sim p(\mathbf{x}_i|\mathbf{x}_o)}\left[ D_{KL}(q(\mathbf{z}|\mathbf{x}_i, \mathbf{x}_o)||q(\mathbf{z}|\mathbf{x}_o)) \right] \\ & - \mathbb{E}_{\mathbf{x}_i \sim p(\mathbf{x}_i|\mathbf{x}_o)}\mathbb{E}_{\mathbf{x}_\phi \sim p(\mathbf{x}_\phi|\mathbf{x}_i, \mathbf{x}_o)}\left[ D_{KL}(q(\mathbf{z}|\mathbf{x}_\phi, \mathbf{x}_i, \mathbf{x}_o)||q(\mathbf{z}|\mathbf{x}_\phi, \mathbf{x}_o)) \right] \\ = {} & \mathbb{E}_{\mathbf{x}_i \sim p(\mathbf{x}_i|\mathbf{x}_o)}\left[ D_{KL}(q(\mathbf{z}|\mathbf{x}_i, \mathbf{x}_o)||q(\mathbf{z}|\mathbf{x}_o)) \right] \\ & - \mathbb{E}_{\mathbf{x}_\phi, \mathbf{x}_i \sim p(\mathbf{x}_\phi, \mathbf{x}_i|\mathbf{x}_o)}\left[ D_{KL}(q(\mathbf{z}|\mathbf{x}_\phi, \mathbf{x}_i, \mathbf{x}_o)||q(\mathbf{z}|\mathbf{x}_\phi, \mathbf{x}_o)) \right] = \hat{R}(i, \mathbf{x}_o). \end{aligned}$$

This new objective tries to maximize the shift of belief on latent variables $\mathbf{z}$ by introducing $\mathbf{x}_i$, while penalizing the information that cannot be absorbed by $\mathbf{x}_\phi$ (by the penalty term $D_{KL}(q(\mathbf{z}|\mathbf{x}_\phi, \mathbf{x}_i, \mathbf{x}_o)||q(\mathbf{z}|\mathbf{x}_\phi, \mathbf{x}_o)))$. Moreover, it is more computationally efficient since one set of samples $\mathbf{x}_\phi, \mathbf{x}_i \sim p(\mathbf{x}_\phi, \mathbf{x}_i|\mathbf{x}_o)$ can be shared across different terms, and the KL-divergence between common parameterizations of encoder (such as Gaussians and normalizing flows) can be computed exactly without the need for approximate integrals. Note also that under approximation

$$p(\mathbf{z}|\mathbf{x}_\phi, \mathbf{x}_i, \mathbf{x}_o) \approx q(\mathbf{z}|\mathbf{x}_\phi, \mathbf{x}_i, \mathbf{x}_o), \ \ p(\mathbf{z}|\mathbf{x}_i, \mathbf{x}_o) \approx q(\mathbf{z}|\mathbf{x}_i, \mathbf{x}_o), \ \ p(\mathbf{z}|\mathbf{x}_o) \approx q(\mathbf{z}|\mathbf{x}_o)$$

, sampling $\mathbf{x}_i \sim p(\mathbf{x}_i|\mathbf{x}_o)$ is approximated by $\mathbf{x}_i \sim \hat{p}(\mathbf{x}_i|\mathbf{x}_o)$, where $\hat{p}(\mathbf{x}_i|\mathbf{x}_o)$ is defined by the following process in Partial VAE. It is implemented by first sampling $\mathbf{z} \sim q(\mathbf{z}|\mathbf{x}_o)$, and then $\mathbf{x}_i \sim p(\mathbf{x}_i|\mathbf{z})$. The same applies for $p(\mathbf{x}_i, \mathbf{x}_\phi|\mathbf{z})$.

# B    ADDITIONAL EXPERIMENTAL RESULTS

## B.1    IMAGE INPAINTING

### B.1.1    PREPROCESSING AND MODEL DETAILS

For our MNIST experiment, we randomly draw 10% of the whole data to be our test set. Partial VAE models (ZI, ZI-m, PNP and PNs) share the same size of architecture with 20 dimensional diagonal

Gaussian latent variables: the generator (decoder) is a 20-200-500-500 fully connected neural network with ReLU activations (where D is the data dimension, $D = 784$). The inference nets (encoder) share the same structure of D-500-500-200-40 that maps the observed data into distributional parameters of the latent space. For the PN-based parameterizations, we use a 500 dimensional feature mapping $h$ parameterized by a single layer neural network, and 20 dimensional ID vectors $\mathbf{e}_i$ (see Section 3.2) for each variable. We choose the symmetric operator $g$ to be the basic summation operator.

During training, we apply Adam optimization (Kingma & Ba, 2015) with default hyperparameter setting, learning rate of 0.001 and a batch size of 100. We generate partially observed MNIST dataset by adding artificially missingness at random in the training dataset during training. We first draw a missing rate parameter from a uniform distribution $\mathscr{U}(0, 0.7)$ and randomly choose variables as unobserved. This step is repeated at each iteration. We train our models for 3K iterations.

### B.1.2 Image generation of partial VAEs

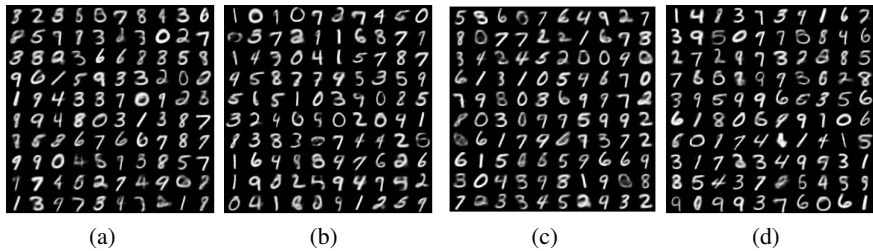

| (a) | (b) | (c) | (d) |

Figure 8: Random images generated using **(a)** naive zero imputing, **(b)** zero imputing with mask, **(c)** PN and **(d)** PNP, respectively.

## B.2 UCI datasets

We applied EDDI on 6 UCI datasets; Boston Housing, Concrete compressive strength, energy efficiency, wine quality, Kin8nm, and Yacht Hydrodynamics. The variables of interest $\mathbf{x}_\phi$ are chosen to be the target variables of each UCI dataset in the experiment.

### B.2.1 Preprocessing and model details

All data are normalized and then scaled between 0 and 1. For each of the 10 - in total- repetitions, we randomly draw 10% of the data to be our test set. Partial VAE models (ZI, ZI-m, PNP and PNs) share the same size of architecture with 10 dimensional diagonal Gaussian latent variables: the generator (decoder) is a 10-50-100-D neural network with ReLU activations (where D is the data dimensions). The inference nets (encoder) share the same structure D-100-50-20 that maps the observed data into distributional parameters of the latent space. For the PN-based parameterizations, we further use a 20 dimensional feature mapping $h$ parameterized by a single layer neural network and 10 dimensional ID vectors $\mathbf{e}_i$ (please refer to section 3.2) for each variable. We choose the symmetric operator $g$ to be the basic summation operator.

As in the image inpainting experiment, we apply Adam optimization during training with default hyperparameter setting, and a batch size of 100 and ingest random missingness as before. We trained our models for 3K iterations.

During active learning, we draw 50 samples in order to estimate the expectation under $\mathbf{x}_\phi, \mathbf{x}_i \sim p(\mathbf{x}_\phi, \mathbf{x}_i | \mathbf{x}_o)$ in Equation (8). Negative likelihoods of the target variable is also estimated using 50 samples of $\mathbf{x}_\phi \sim p(\mathbf{x}_\phi | \mathbf{x}_o)$ through $p(\mathbf{x}_\phi | \mathbf{x}_o) \approx \frac{1}{M} \sum_{m=1}^{M} p(\mathbf{x}_\phi | \mathbf{z}_m)$, where $\mathbf{z}_m \sim q(\mathbf{z} | \mathbf{x}_o)$.

### B.2.2 Tables on Area under the information curve (AUIC)

In addition to the area under the information curve (AUIC) ranking metric provided in the paper, the average area under the information curve (Avg. AUIC) on each dataset can also be used to compare the performance across models and strategies. AUIC is defined to be $\sum_t -\log p(\mathbf{x}_\phi | \mathbf{x}_{O_t})$, where $\mathbf{x}_{O_t}$ is the basket of variables observed at step $t$. By definition, smaller AUIC value (could be positive or

negative) indicates better performance. We present the AUIC for each dataset in Table 6, 7, 8, 9, 10, and 11 [2].

Readers might have found that it seems that the avg. AUIC results in Tables 6 - 11 contradicts the avg. ranking of AUIC results in Table 2 of the main text. However, this is not the case. In Tables Tables 6 - 11, AUIC numbers only provide a simplified statistics of *marginal* distributions of each method's performance. Here, the distribution of performance is defined by first sample a data point from the data distribution, and then we obtain the performance of a active learning method of interest by evaluating its performance (AUIC) on this single data point. On the contrary, the average AUIC *ranking* measure actually takes into account the *joint distributions* of the performance of all methods, since ranking is a function of the performance of all methods.

With this additional information of correlations, this gives a more accurate evaluation regarding the actual performance of different methods. Notably, in practical scenario of active variable selection, the latter setting is obviously more sensible and fare.

The above conjecture is further validated and confirmed by applying the nonparametric statistical test, namely the Wilcoxcon signed-rank significance test on the performance of different methods, which are detailed in Appendix B.2.3. Wilcoxon test is a very powerful statistical test which includes the information of the joint distribution in paired samples. In our case, the term *paired samples* refers to the situation that different algorithms are evaluated on exactly the same set of test data points, which introduces correlations between the performances of different algorithms.

Table 6: Average AUIC over Boston Housing dataset

| Method | ZI | ZI-m | PNP | PN(5) | PN(1) ) |
|--------|-----|------|-----|-------|---------|
| EDDI | **-25.03** (0.09 ) | -24.74 (0.15 ) | -24.49 (0.24 ) | -24.54 (0.10 ) | -24.41 (0.09) |
| Random | -23.85 (0.14 ) | -24.52 (0.08 ) | -23.36 (0.18 ) | -23.43 (0.14 ) | -23.33 (0.13) |
| Single best | -24.77 (0.12 ) | -23.62 (0.20 ) | -23.71 (0.15 ) | -23.82 (0.13 ) | -23.87 (0.09) |

Table 7: Average AUIC over Concrete dataset

| Method | ZI | ZI-m | PNP | PN(5) | PN(1) |
|--------|-----|------|-----|-------|-------|
| EDDI | -12.07 (0.04 ) | -12.07 (0.05 ) | -12.09 (0.07 ) | -12.15 (0.06 ) | **-12.17** (0.07) |
| Random | -11.00 (0.09 ) | -12.03 (0.03 ) | -11.17. (0.07 ) | -12.07 (0.06 ) | -11.12 (0.12) |
| Single best | -12.03 (0.06 ) | -11.13 (0.10 ) | -12.07 (0.06 ) | -12.11 (0.04 ) | -12.16 (0.06) |

Table 8: Average AUIC over Energy dataset

| Method | ZI | ZI-m | PNP | PN(5) | PN(1) |
|--------|-----|------|-----|-------|-------|
| EDDI | -13.63 (0.06 ) | -14.56 (0.07 ) | -14.49 (0.06 ) | -14.65 (0.08 ) | **-14.68** (0.07) |
| Random | -9.89 (0.15 ) | -14.53 (0.06 ) | -11.49. (0.16 ) | -11.67 (0.17 ) | -11.51 (0.16) |
| Single best | -12.79 (0.07 ) | -11.62 (0.08 ) | -14.36 (0.09 ) | -14.56 (0.08 ) | -14.66 (0.07) |

Table 9: Average AUIC over Wine dataset

| Method | ZI | ZI-m | PNP | PN(5) | PN(1) |
|--------|-----|------|-----|-------|-------|
| EDDI | -14.24 (0.06 ) | -15.04 (0.02 ) | -15.04 (0.05 ) | **-15.13** (0.05 ) | -15.10 (0.03) |
| Random | -11.07 (0.20 ) | -15.10 (0.05 ) | -12.38 (0.9 ) | -12.57 (0.14 ) | -12.26 (0.15) |
| Single best | -13.85 (0.10 ) | -12.55 (0.10 ) | -15.02 (0.05 ) | -14.99 (0.03 ) | -15.04 (0.03) |

---

[2]Note that the notation of PN(5) indicates an extension of the PN-Partial VAE method which will be discussed in detail later in the Appendix B.5.

Table 10: Average AUIC over kin8nm dataset

| Method | ZI | ZI-m | PNP | PN(5) | PN(1) |
|---|---|---|---|---|---|
| EDDI | **-20.31** (0.02 ) | -20.25 (0.01 ) | -20.18 (0.04 ) | -20.20 (0.02 ) | -20.15 (0.02) |
| Random | -19.40 (0.04 ) | -20.24 (0.02 ) | -19.29 (0.04 ) | -19.41 (0.02 ) | -19.28 (0.05) |
| Single best | -20.28 (0.02 ) | -19.35 (0.04 ) | -20.23 (0.03 ) | -20.19 (0.01 ) | -20.19 (0.03) |

Table 11: Average AUIC over Yacht dataset

| Method | ZI | ZI-m | PNP | PN(5) | PN(1) |
|---|---|---|---|---|---|
| EDDI | -14.37 (0.02 ) | -14.50 (0.02 ) | **-14.57** (0.02 ) | -14.53 (0.02) | -14.56 (0.02) |
| Random | -12.83 (0.03 ) | -14.50 (0.02 ) | -13.03 (0.04 ) | -12.93 (0.04 ) | -13.08 (0.02) |
| Single best | -14.43 (0.03 ) | -12.91 (0.03 ) | **-14.57** (0.02 ) | -14.50 (0.03 ) | -14.54(0.02) |

### B.2.3 STATISTICAL SIGNIFCANT TEST RESULTS

In this section, we perform Wilcoxcon signed-rank significance test on the performance of different methods, to support our result in Table 2. Since Table 2 suggests that EDDI-PNP-Partial VAE is the best algorithm overall, we set EDDI-PNP-Partial VAE as default and perform Wilcoxcon test between EDDI-PNP-Partial VAE and all other 15 different settings, to see whether the improvement is significant. Table 12 displays the corresponding p-value for each test. It is obvious that in all 15 tests, the EDDI-PNP-Partial VAE results are significant (compared with the standard $\alpha = 0.05$ cutoff). This provides strong evidence that confirms our results in Table 2 and our conjecture in Appendix B.2.2.

Table 12: p- values of Wilcoxon signed-rank test of EDDI-PNP vs. 11 other settings, on 6 UCI datasets.

| Method | ZI | ZI-m | PNP | PN |
|---|---|---|---|---|
| EDDI | $< 10^{-48}$ | $<10^{-23}$ | N/A | $<10^{-2}$ |
| Random | 0 | 0 | 0 | 0 |
| Single best | 0 | $<10^{-13}$ | $<10^{-2}$ | $<10^{-4}$ |

### B.2.4 ADDITIONAL AUIC PLOTS OF PN, ZI AND ZI-M ON UCI DATASETS

Here we present additional plots of the information curve during active variable selection. Figure 9 presents the results for the Boston Housing, the Energy and the Wine datasets and for the three approaches, i.e. PN, ZI and masked ZI.

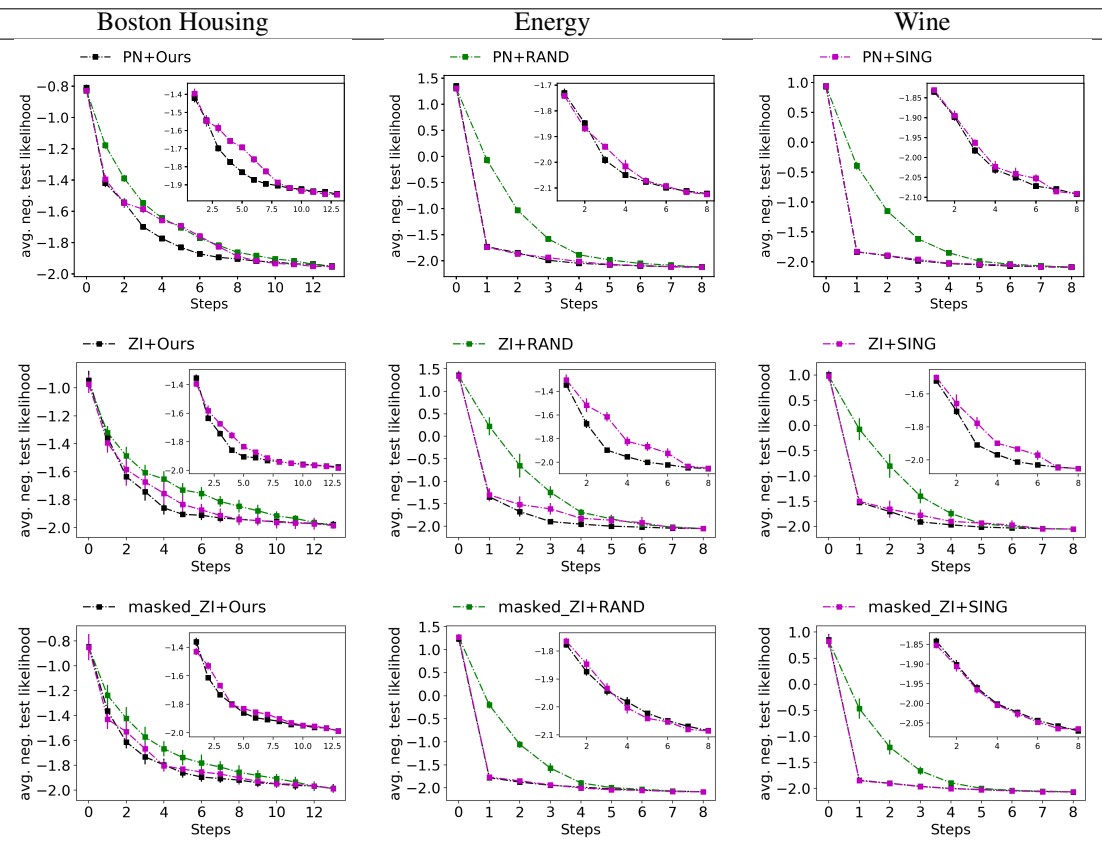

Figure 9: Information curves of active variable selection for the three UCI datasets and the three approaches, i.e. **(First row)** PointNet (PN), **(Second row)** Zero Imputing (ZI), and **(Third row)** Zero Imputing with mask (ZI-m). **Green**: random strategy; **Black**: EDDI; **Pink**: Single best ordering. This displays negative test log likelihood (y axis, the lower the better) during the course of active selection (x-axis).

### B.2.5    RMSE PLOTS OF PN, ZI AND ZI-M ON UCI DATASETS

Here we present additional plots of the RMSE curves during active learning. Figure 10 presents the results for the Boston Housing, the Energy and the Wine datasets and for the three approaches, i.e. PN, ZI and masked ZI.

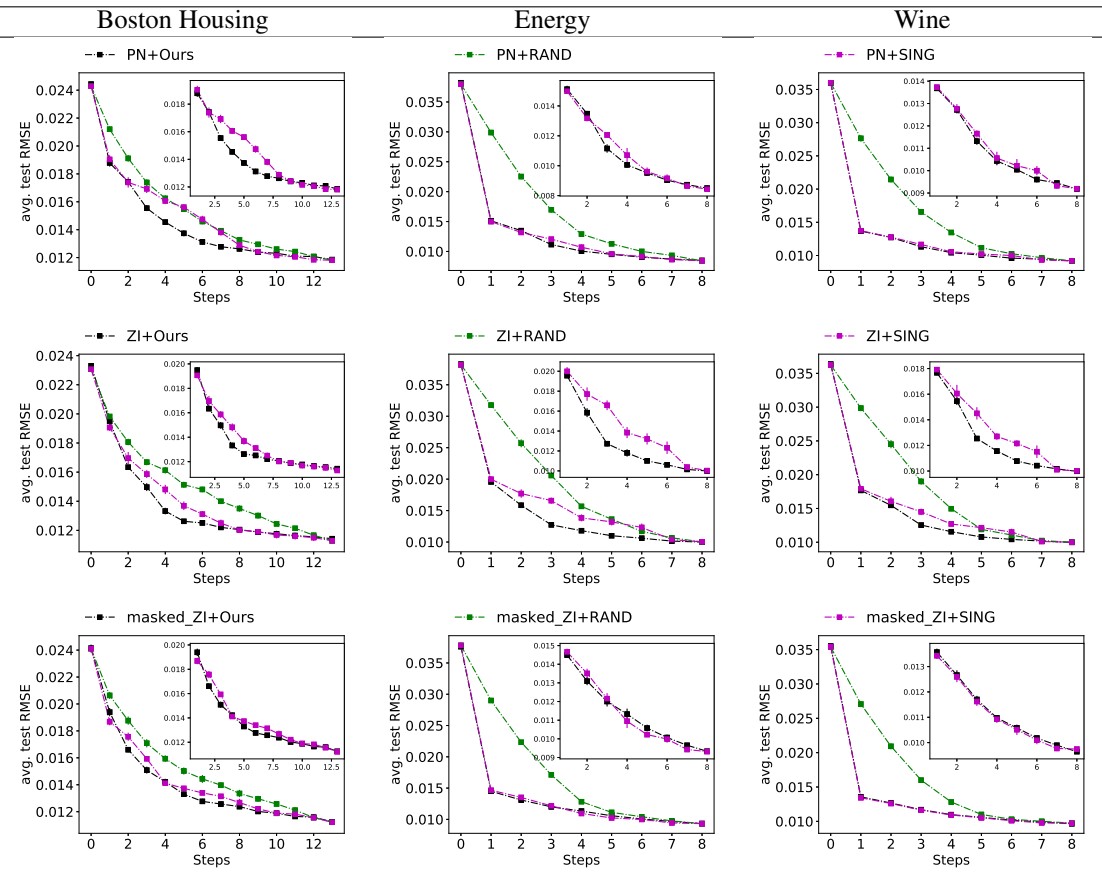

Figure 10: RMSE curves of active variable selection for the three UCI datasets and the three approaches, i.e. **(First row)** PointNet (PN), **(Second row)** Zero Imputing (ZI), and **(Third row)** Zero Imputing with mask (ZI-m). **Green**: random strategy; **Black**: EDDI; **Pink**: Single best ordering. This displays RMSE (y axis, the lower the better) during the course of active selection (x-axis).

### B.2.6 COMPARISONS BETWEEN EDDI AND LASSO-BASED METHOD

Here we present additional results of a new baseline, the LASSO-based feature selection. This is not presented in the main text since LASSO is designed for a different problem setting. It requires fully observed data, and only works in regression problems with one dimensional outputs. Both MIMIC III and NHANES tasks do not fulfill these requirements. Additionally, LASSO aims to select a global set of features to obtain the best performance instead of select the most informative feature given partially observed information, thus cannot be used in a sequential setting. We thus construct the LASSO feature selection baseline as follows for comparison: we first apply LASSO regression on training dataset which is fully observed in these UCI datasets, and select the features (denoted by $\mathscr{A}$) that correspond to non-zero coefficients. Then, during test time, LASSO strategy will observe the features one by one from $\mathscr{A}$ randomly. When all variables selected by LASSO are already picked, we stop the feature selection progress. Since LASSO does not support evaluation of model likelihood as well as it is linear, we use the corresponding partial-VAE (ZI,ZI-m,PNP,PN) to make predictions and evaluate the model log likelihood.

Figure 11 presents the results for the Boston Housing, the Energy and the Wine datasets as examples. Full results of all UCI datasets are presented in Table 13. Note that in Table 13, Wilcoxon signed-rank test is performed between EDDI and LASSO strategies for each Partial VAE models, respectively. The results indicates that EDDI significantly outperforms LASSO in all circumstances. This is despite the fact that EDDI is a greedy sequential variable selection method that built upon partially observed data, while LASSO-baseline makes use of the information from *fully observed data*, and selects the

set of variables in a *non-greedy, global manner*, which is often unrealistic in many pratical application settings.

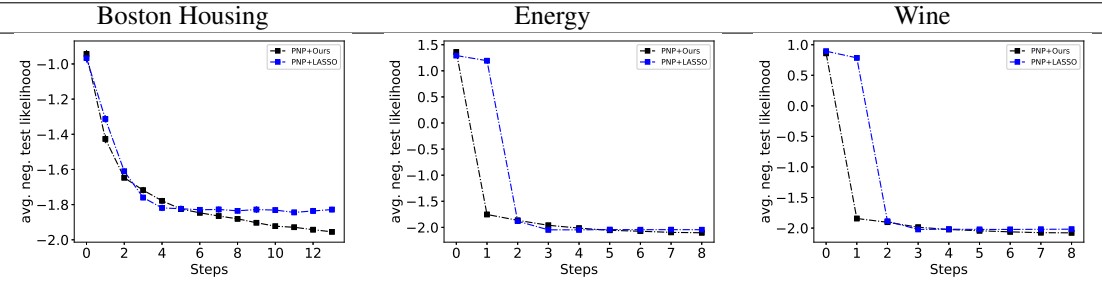

Figure 11: Information curves of active variable selection for the three UCI datasets and PNP-Partial VAE. **Black**: EDDI; **Blue**: Single best ordering. This displays negative test log likelihood (y axis, the lower the better) during the course of active selection (x-axis).

Table 13: Avg. rankings of AUIC, and p- values of Wilcoxon signed-rank test that EDDI outperforms LASSO (on 6 UCI datasets).

| Method | ZI | ZI-m | PNP | PN |
|---|---|---|---|---|
| EDDI | 4.66 (0.02) | 4.53(0.02) | **4.14**(0.02) | 4.24(0.02) |
| LASSO | 4.86(0.02) | 4.63(0.02) | 4.41(0.02) | 4.48(0.02) |
| p-value | $<10^{-4}$ | $<10^{-6}$ | $<10^{-24}$ | $<10^{-19}$ |

### B.2.7   ILLUSTRATION OF DECISION PROCESS OF EDDI (BOSTON HOUSING AS EXAMPLE)

The decision process facilitated by the active selection of the variables (for the EDDI framework) is efficiently illustrated in Figure 12 and Figure 13 for the Boston Housing dataset and for the PNP and PNP with single best ordering approaches, respectively.

For completeness, we provide details regarding the abbreviations of the variables used in the Boston dataset and appear both figures.

CR - per capita crime rate by town

PRD - proportion of residential land zoned for lots over 25,000 sq.ft.

PNB - proportion of non-retail business acres per town.

CHR - Charles River dummy variable (1 if tract bounds river; 0 otherwise)

NOC - nitric oxides concentration (parts per 10 million)

ANR - average number of rooms per dwelling

AOUB - proportion of owner-occupied units built prior to 1940

DTB - weighted distances to five Boston employment centres

ARH - index of accessibility to radial highways

TAX - full-value property-tax rate per $10,000

OTR - pupil-teacher ratio by town

PB - proportion of blacks by town

LSP - % lower status of the population

### B.3   MIMIC-III

Here we provide additional results of our approach on the MIMIC-III dataset.

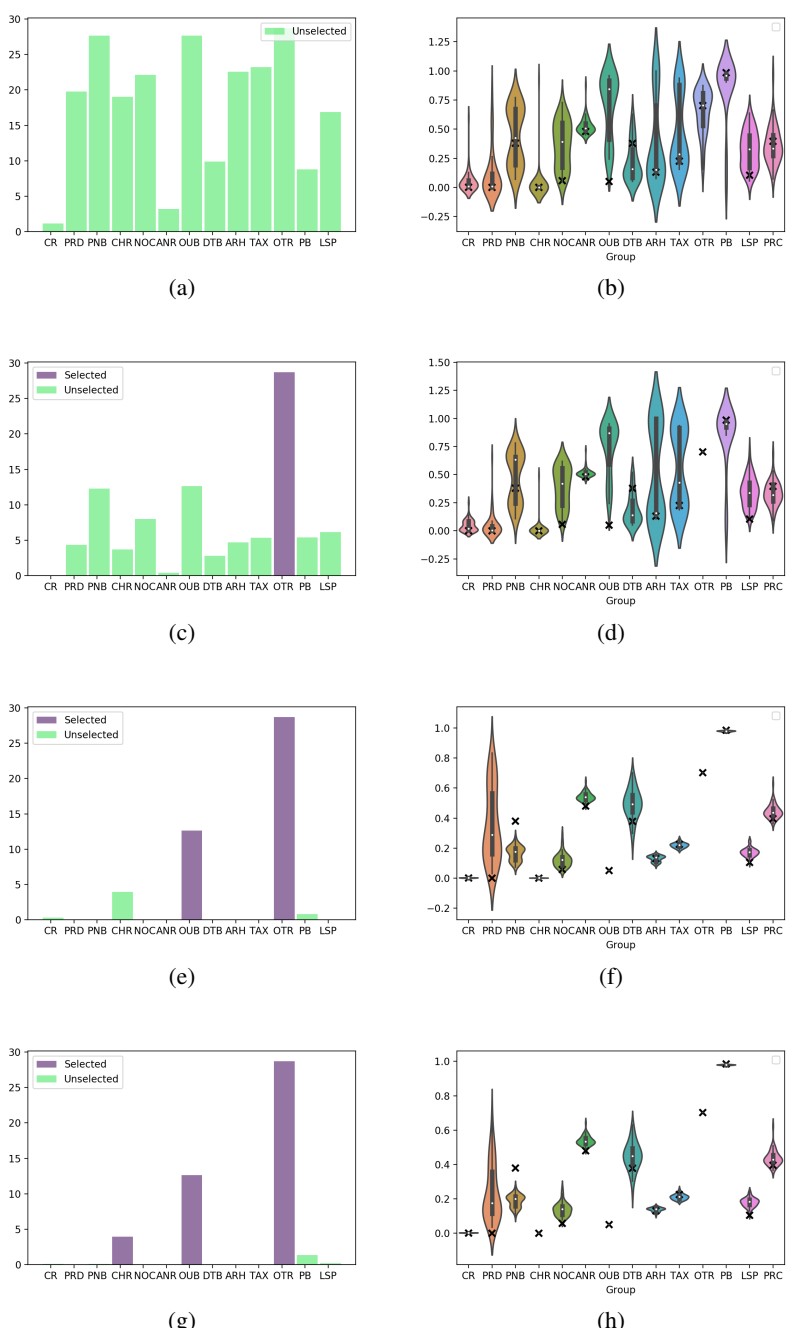

Figure 12: Information reward estimated during the first 4 active variable selection steps on a randomly chosen Boston Housing test data point. **Model**: PNP, strategy: EDDI. Each row contains two plots regarding the same time step. **Bar plots on the left** show the information reward estimation of each variable on the y-axis. All unobserved variables start with green bars, and turns purple once selected by the algorithm. **Right**: violin plot of the posterior density estimations of remaining unobserved variables.

.

### B.3.1 PREPROCESSING AND MODEL DETAILS

For our active learning experiments on MIMIC III datasets, we chose the variable of interest $\mathbf{x}_\phi$ to be the binary mortality indicator of the dataset. All data (except the binary mortality indicator)

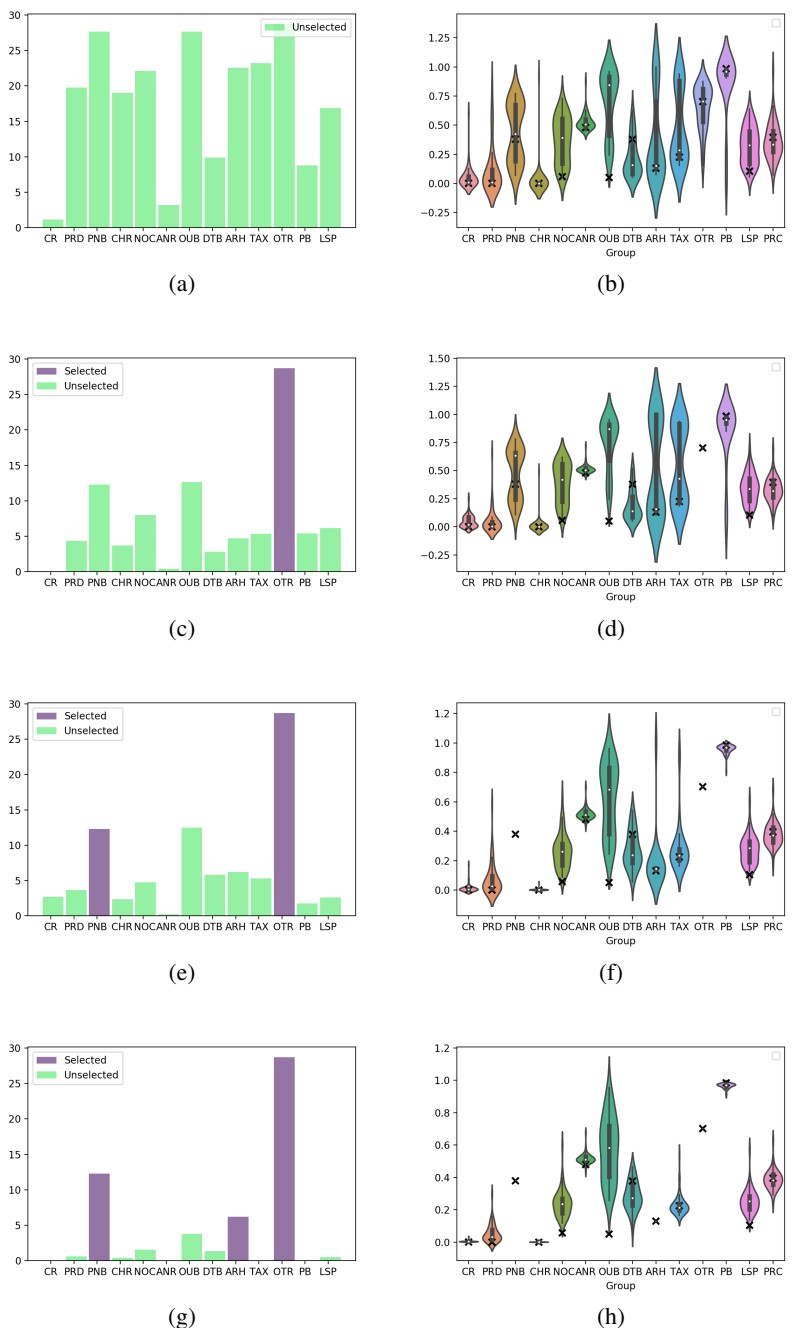

Figure 13: Information reward estimated during the first 4 active variable selection steps on a randomly chosen Boston Housing test data point. **Models**: PNP, strategy: single ordering. Each row contains two plots regarding the same time step. **Bar plots on the left** show the information reward estimation of each variable on the y-axis. All unobserved variables start with green bars, and turns purple once selected by the algorithm. **Right**: violin plot of the posterior density estimations of remaining unobserved variables.

.

are normalized and then scaled between 0 and 1. We transformed the categorical variables into real-valued using the dictionary deduced from (Johnson et al., 2016) that makes use of the actual

medical implications of each possible values. The binary mortality indicator are treated as Bernoulli variables and Bernoulli likelihood function is applied. For each repetition (of the 5 in total), we randomly draw 10% of the whole data to be our test set. Partial VAE models (ZI, ZI-m, PNP and PNs) share the same size of architecture with 10 dimensional diagonal Gaussian latent variables: the generator (decoder) is a 10-50-100-D neural network with ReLU activations (where D is the data dimensions). The inference nets (encoder) share the same structure of D-100-50-20 that maps the observed data into distributional parameters of the latent space. Additionally, for PN-based parameterizations, we further use a 20 dimensional feature mapping $h$ parameterized by a single layer neural network, and 10 dimensional ID vectors $\mathbf{e}_i$ (please refer to section 3.2) for each variable. We choose the symmetric operator $g$ to be the basic summation operator.

Adam optimization and random missingness is applied as in the previous experiments. We trained our models for 3K iterations. During active learning, we draw 50 samples in order to estimate the expectation under $\mathbf{x}_\phi, \mathbf{x}_i \sim p(\mathbf{x}_\phi, \mathbf{x}_i | \mathbf{x}_o)$ in Equation (8). Negative likelihoods of the target variable is also estimated using 50 samples of $\mathbf{x}_\phi \sim p(\mathbf{x}_\phi | \mathbf{x}_o)$ through $p(\mathbf{x}_\phi | \mathbf{x}_o) \approx \frac{1}{M} \sum_{m=1}^{M} p(\mathbf{x}_\phi | \mathbf{z}_m)$, where $\mathbf{z}_m \sim q(\mathbf{z} | \mathbf{x}_o)$.

### B.3.2 ADDITIONAL PLOTS OF ZI, PN AND ZI-M ON MIMIC III

Figure 14 shows the information curves of active variable selection on the risk assessment task for MIMIC-III as produced by the three approaches, i.e. ZI, PN and masked ZI.

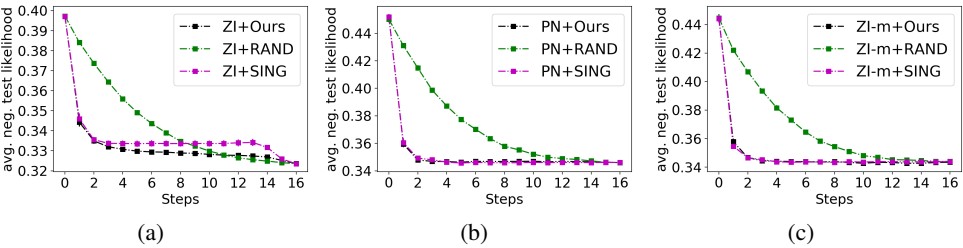

(a)       (b)       (c)

Figure 14: Information curves of active variable selection on risk assessment task on MIMIC III, produced from: **(a)** Zero Imputing (ZI), **(b)** PointNet (PN) and **(c)** Zero Imputing with mask (ZI-m). **Green**: random strategy; **Black**: EDDI; **Pink**: Single best ordering. This displays negative test log likelihood (y axis, the lower the better) during the course of active selection (x-axis)
.

### B.4 NHANES

#### B.4.1 PREPROCESSING AND MODEL DETAILS

For our active learning experiments on NHANES datasets, we chose the variable of interest $\mathbf{x}_\phi$ to be the lab test result section of the dataset. All data are normalized and scaled between 0 and 1. For categorical variables, these are transformed into real-valued variables using the code that comes with the dataset, which makes use of the actual ordering of variables in questionnaire. Then, for each repetition (of the 5 repetitions in total), we randomly draw 8000 data as training set and 100 data to be test set. All partial VAE models (ZI, ZI-m, PNP and PNs) uses gaussian likelihoods, with an diagonal Gaussian inference model (encoder). Partial VAE models share the same size of architecture with 20 dimensional diagonal Gaussian latent variables: the generator (decoder) is a 20-50-100-D neural network. The inference nets (encoder) share the same structure of D-100-50-20 that maps the observed data into distributional parameters of the latent space. Additionally, for PN-based parameterizations, we further use a 20 dimensional feature mapping $h$ parameterized by a single layer neural network, and 100 dimensional ID vectors $\mathbf{e}_i$ (please refer to section 3.2) for each variable. We choose the symmetric operator $g$ to be the basic summation operator.

Adam optimization and random missingness is applied as in the previous experiments. We trained all models 1K iterations. During active learning, 10 samples were drawn to estimate the expectation in Equation (9). Negative likelihoods of the target variable is also estimated using 10 samples.

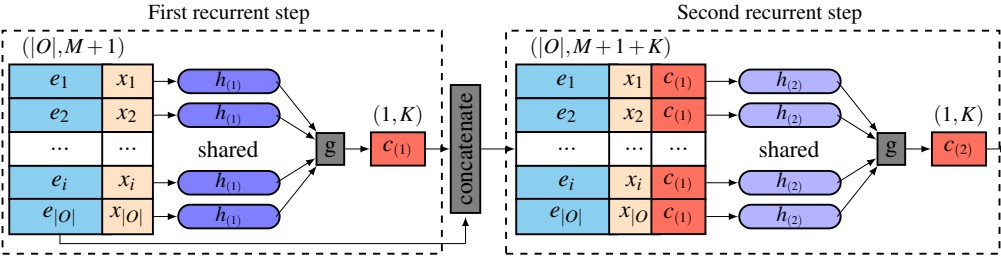

Figure 15: Illustration of recurrent PN architecture. We show the example using 2 recurrent steps. The output $c_{(2)}$ is directly connected to the rest of the inference network in this case. One can use more steps. To form the input for the $i+1$ recurrent step, we concatenate the $c_{(i)}$ to the input.

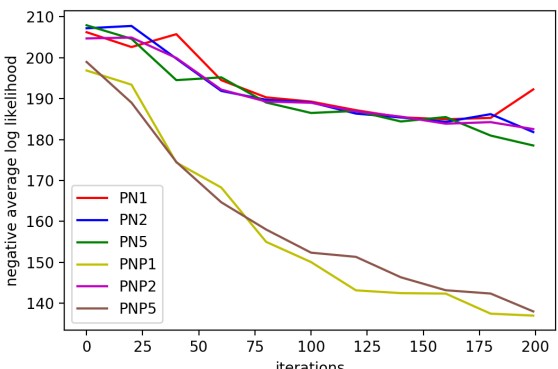

Figure 16: Negative test log likelihoods of pilot runs for (recurrent) PN-based methods on MNIST dataset. we perform plot runs of PN1, PN2, PN5, PNP1, PNP1, PNP2, PNP5 (PN-x stands for x recurrent steps of PN) on MNIST dataset for 300 iterations. All curves has been smoothed for clear comparison.

### B.5 PN/PNP MODEL STRUCTURE DETERMINATION: SHOULD WE USE RECURRENT EXTENSIONS

One straightforward extention of PN/PNP Partial VAEs proposed in this paper is, to generalize PN and PNP by recurrently reuse the code $c$ to enlarge the capacity of PN:

Figure 15 shows the mechanism of the recurrent PN with two recurrent steps using concatenated $\mathbf{s}_{(1)d} = [\mathbf{e}_d, \mathbf{x}_d]$ as an example. The first step is the same as the PN setting with the $K$ dimensional output $c_{(1)}$, where $_{(1)}$ is the recurrent step index. For the second step, we concatenate the learned $c_{(1)}$ to $\mathbf{s}_{(1)}$ to form the new input for the next recurrent step $\mathbf{s}_{(2)d} = [\mathbf{s}_{(1)d}, c_{(1)}]$. There can be arbitrary number of recurrent steps using the input $\mathbf{s}_{(n)d} = [\mathbf{s}_{(n-1)d}, c_{(n-1)}]$. Within each recurrent step, the parameters for the neural network $h_{(n)}$ are shared, however, different steps have different parameters. When $n = 1$, we recover the original PN partial VAE setting.

The question is, should we include the recurrent structure in our Partial VAE? In this section, we present preliminary result for this purpose. We perform plot runs of PN1, PN2, PN5, PNP1, PNP1, PNP2, PNP5 (PN-x stands for PN model with x recurrent steps) on MNIST dataset for 300 iterations. Other model settings are consistent with Section B.1.1. Results of negative test log likelihoods are shown in Figure 16: Based on Figure 16, the conclusion is: based on MNIST dataset along, by increasing the recurrent steps of PN, the performance roughly increases slightly. Meanwhile, we can observe that increasing the recurrent steps does not improve PNP. However, the difference is not significant.

In section B.2.2, a comparison between recurrent PN (PN5) and vanilla PN (PN1) is considered for each UCI dataset, see Table 6, 7, 8, 9, 10, and 11. Additionally, the average ranking of AUIC of PN5 and PN1 is summarized in Table 14:

Table 14: Average Ranking of AUIC between PN5-Partial VAE and PN1-Partial VAE

| Method | PN(5) | PN(1) |
|---|---|---|
| EDDI | 2.83 (0.01 ) | 2.78 (0.01 ) |
| Random | 4.12 (0.01 ) | 4.07 (0.01 ) |
| Single best | 4.37 (0.01 ) | 2.80 (0.01 ) |

It is clear that one can *not* conclude that on average, PN5 significantly outperforms PN1. Therefore, as far as active learning tasks under the settings in our experiments are considered, we believe the recurrent generalization of PNs will not be a crucial factor for boosting a performance.

## C   ADDITIONAL THEORETICAL CONTRIBUTIONS

### C.1   ZERO IMPUTING AS A POINT NET

Here we present how the zero imputing (ZI) and PointNet (PN) approaches relate.

**Zero imputation with inference net** In ZI, the natural parameter of $\lambda$ (e.g., Gaussian parameters in variational autoencoders) is approximated using the following neural network:

$$f(\mathbf{x}) := \sum_{l=1}^{L} w_l^{(1)} \sigma(\mathbf{w}_l^{(0)} \mathbf{x}^T)$$

,

where $L$ is the number of hidden units, $\mathbf{x}$ is the input image with $x_i$ be the value of the $i^{th}$ pixel. To deal with partially observed data $\mathbf{x} = \mathbf{x}_o \cup \mathbf{x}_u$, ZI simply sets all $\mathbf{x}_u$ to zero, and use the full inference model $f(\mathbf{x})$ to perform approximate inference.

**PointNet parameterization** The PN approach approximates the natural parameter $\lambda$ by a permutation invariant set function

$$g(h(\mathbf{s}_1), h(\mathbf{s}_2), ..., h(\mathbf{s}_O)),$$

where $\mathbf{s}_i = [x_i, \mathbf{e}_i]$, $\mathbf{e}_i$ is the $I$ dimensional embedding/ID/location vector of the $i^{th}$ pixel, $g(\cdot)$ is a symmetric operation such as max-pooling and summation, and $h(\cdot)$ is a nonlinear feature mapping from $\mathbb{R}^{I+1}$ to $\mathbb{R}^K$ (we will always refer $h$ as *feature maps* ). In the current version of the partial-VAE implementation, where Gaussian approximation is used, we set $K = 2H$ with $H$ being the dimension of latent variables. We set $g$ to be the element-wise summation operator, i.e. a mapping from $\mathbb{R}^{KO}$ to $\mathbb{R}^K$ defined by:

$$g(h(\mathbf{s}_1), h(\mathbf{s}_2), ..., h(\mathbf{s}_O)) = \sum_{i \in O} h(\mathbf{s}_i).$$

This parameterization corresponds to products of multiple Exp-Fam factors $\prod_{i \in O} \exp\{-\langle h(\mathbf{s}_i), \Phi \rangle\}$.

**From PN to ZI** To derive the PN correspondence of the above ZI network we define the following PN functions:

$$h(\mathbf{s}_i) := \mathbf{e}_i * x_i$$

$$g(h(\mathbf{s}_1), h(\mathbf{s}_2), ..., h(\mathbf{s}_O)) := \sum_{k=1}^{I} \theta_k \sigma(\sum_{i \in O} h_k(\mathbf{s}_i)),$$

where $h_k(\cdot)$ is the $k^{th}$ output feature of $h(\cdot)$. The above PN parameterization is also permutation invariant; setting $L = I$, $\theta_l = w_l^{(1)}$, $(\mathbf{w}_l^{(0)})_i = (\mathbf{e}_i)_l$ the resulting PN model is equivalent to the ZI neural network.

**Generalizing ZI from PN perspective** In the ZI approach, the missing values are replaced with zeros. However, this ad-hoc approach does not distinguish missing values from actual observed zero values. In practice, being able to distinguish between these two is crucial for improving uncertainty estimation during partial inference. One the other hand, we have found that PN-based partial VAE experiences difficulties in training. To alleviate both issues, we proposed a generalization of the ZI approach that follows a PN perspective. One of the advantages of PN is setting the *feature maps* of the unobserved variables to zero instead of the related weights. As discussed before, these two approaches

are equivalent to each other only if the factors are linear. More generally, we can parameterize the PN by:

$$h^{(1)}(\mathbf{s}_i) := \mathbf{e}_i * x_i$$
$$h^{(2)}(h_i^{(1)}) := NN_1(h_i^{(1)})$$
$$g(h(\mathbf{s}_1), h(\mathbf{s}_2), ..., h(\mathbf{s}_O)) := NN_2(\sigma(\sum_{i \in O} h_k^{(2)}(h_i^{(1)}))),$$

where $NN_1$ is a mapping from $\mathbb{R}^I$ to $\mathbb{R}^K$ defined by a neural network, and $NN_2$ is a mapping from $\mathbb{R}^K$ to $\mathbb{R}^{2H}$ defined by another neural network.

## C.2 Approximation Difficulty of the Acquisition Function

Traditional variational approximation approaches provide wrong approximation direction when applied in this case (resulting in an upper bound of the objective $R_\phi(i, \mathbf{x}_O)$ which we maximize). Justification issues aside, (black box) variational approximation requires sampling from approximate posterior $q(\mathbf{z}|\mathbf{x}_O)$, which leads to extra uncertainties and computations. For common proposals of approximation:

- Directly estimate entropy via sampling $\Rightarrow$ problematic for high dimensional target variables
- Using reversed information reward $\mathbb{E}_{\mathbf{x}_i \sim p(\mathbf{x}_i|\mathbf{x}_o)}[D_{KL}(p(\mathbf{x}_\phi|\mathbf{x}_o)||p(\mathbf{x}_\phi|\mathbf{x}_o, \mathbf{x}_i))]$, and then apply ELBO (KL-divergence) $\Rightarrow$ This does not make sense mathematically, since this will result in upper bound approximation of the (reversed) information objective, this is in the wrong direction.
- Ranganath's bound (Ranganath et al., 2016) on estimating entropy$\Rightarrow$ gives upper bound of the objective, wrong direction.
- All the above methods also needs samples from latent space (therefore second level approximation needed).

## C.3 Connection of EDDI information reward with BALD

We briefly discuss connection of EDDI information reward with BALD (Houlsby et al., 2011) and. MacKay's work (MacKay, 1992). Assuming the model is correct, i.e. $q = p$, we have

$$R(i, \mathbf{x}_o) = \mathbb{E}_{\mathbf{x}_i \sim p(\mathbf{x}_i|\mathbf{x}_o)}\left[D_{KL}(p(\mathbf{z}|\mathbf{x}_i, \mathbf{x}_o)||p(\mathbf{z}|\mathbf{x}_o))\right]$$
$$- \mathbb{E}_{\mathbf{x}_i \sim p(\mathbf{x}_i|\mathbf{x}_o)}\mathbb{E}_{\mathbf{x}_\phi \sim p(\mathbf{x}_\phi|\mathbf{x}_i, \mathbf{x}_o)}\left[D_{KL}(p(\mathbf{z}|\mathbf{x}_\phi, \mathbf{x}_i, \mathbf{x}_o)||p(\mathbf{z}|\mathbf{x}_\phi, \mathbf{x}_o))\right].$$

Note that based on McKay's relationship between entropy and KL-divergence reduction, we have:

$$\mathbb{E}_{\mathbf{x}_i \sim p(\mathbf{x}_i|\mathbf{x}_o)}\left[D_{KL}(p(\mathbf{z}|\mathbf{x}_i, \mathbf{x}_o)||p(\mathbf{z}|\mathbf{x}_o))\right]$$
$$= \mathbb{E}_{\mathbf{x}_i \sim p(\mathbf{x}_i|\mathbf{x}_o)}\left[H(p(\mathbf{z}|\mathbf{x}_i, \mathbf{x}_o)) - H(p(\mathbf{z}|\mathbf{x}_o))\right].$$

Similarly, we have

$$\mathbb{E}_{\mathbf{x}_i \sim p(\mathbf{x}_i|\mathbf{x}_o)}\mathbb{E}_{\mathbf{x}_\phi \sim p(\mathbf{x}_\phi|\mathbf{x}_i, \mathbf{x}_o)}\left[D_{KL}(p(\mathbf{z}|\mathbf{x}_\phi, \mathbf{x}_i, \mathbf{x}_o)||p(\mathbf{z}|\mathbf{x}_\phi, \mathbf{x}_o))\right]$$
$$= \mathbb{E}_{\mathbf{x}_\phi \sim p(\mathbf{x}_\phi|\mathbf{x}_o)}\mathbb{E}_{\mathbf{x}_i \sim p(\mathbf{x}_i|\mathbf{x}_\phi, \mathbf{x}_o)}\left[D_{KL}(p(\mathbf{z}|\mathbf{x}_\phi, \mathbf{x}_i, \mathbf{x}_o)||p(\mathbf{z}|\mathbf{x}_\phi, \mathbf{x}_o))\right]$$
$$= \mathbb{E}_{\mathbf{x}_\phi \sim p(\mathbf{x}_\phi|\mathbf{x}_o)}\mathbb{E}_{\mathbf{x}_i \sim p(\mathbf{x}_i|\mathbf{x}_\phi, \mathbf{x}_o)}\left[H(p(\mathbf{z}|\mathbf{x}_\phi, \mathbf{x}_i, \mathbf{x}_o)) - H(p(\mathbf{z}|\mathbf{x}_\phi, \mathbf{x}_o))\right]$$
$$= \mathbb{E}_{\mathbf{x}_i \sim p(\mathbf{x}_i|\mathbf{x}_o)}\mathbb{E}_{\mathbf{x}_\phi \sim p(\mathbf{x}_\phi|\mathbf{x}_i, \mathbf{x}_o)}\left[H(p(\mathbf{z}|\mathbf{x}_\phi, \mathbf{x}_i, \mathbf{x}_o))\right] - \mathbb{E}_{\mathbf{x}_\phi \sim p(\mathbf{x}_\phi|\mathbf{x}_o)}\mathbb{E}_{\mathbf{x}_i \sim p(\mathbf{x}_i|\mathbf{x}_\phi, \mathbf{x}_o)}\left[H(p(\mathbf{z}|\mathbf{x}_\phi, \mathbf{x}_o))\right]$$
$$= \mathbb{E}_{\mathbf{x}_i \sim p(\mathbf{x}_i|\mathbf{x}_o)}\mathbb{E}_{\mathbf{x}_\phi \sim p(\mathbf{x}_\phi|\mathbf{x}_i, \mathbf{x}_o)}\left[H(p(\mathbf{z}|\mathbf{x}_\phi, \mathbf{x}_i, \mathbf{x}_o))\right] - \mathbb{E}_{\mathbf{x}_\phi \sim p(\mathbf{x}_\phi|\mathbf{x}_o)}\left[H(p(\mathbf{z}|\mathbf{x}_\phi, \mathbf{x}_o))\right],$$

where MacKay's result is applied to $\mathbb{E}_{\mathbf{x}_i \sim p(\mathbf{x}_i|\mathbf{x}_\phi, \mathbf{x}_o)}\left[D_{KL}(p(\mathbf{z}|\mathbf{x}_\phi, \mathbf{x}_i, \mathbf{x}_o)||p(\mathbf{z}|\mathbf{x}_\phi, \mathbf{x}_o))\right]$. Putting everything together, we have

$$R(i, \mathbf{x}_o) = \mathbb{E}_{\mathbf{x}_i \sim p(\mathbf{x}_i|\mathbf{x}_o)}\left[H(p(\mathbf{z}|\mathbf{x}_i, \mathbf{x}_o)) - H(p(\mathbf{z}|\mathbf{x}_o))\right]$$
$$- \mathbb{E}_{\mathbf{x}_i \sim p(\mathbf{x}_i|\mathbf{x}_o)}\mathbb{E}_{\mathbf{x}_\phi \sim p(\mathbf{x}_\phi|\mathbf{x}_i, \mathbf{x}_o)}\left[H(p(\mathbf{z}|\mathbf{x}_\phi, \mathbf{x}_i, \mathbf{x}_o))\right] + \mathbb{E}_{\mathbf{x}_\phi \sim p(\mathbf{x}_\phi|\mathbf{x}_o)}\left[H(p(\mathbf{z}|\mathbf{x}_\phi, \mathbf{x}_o))\right]$$
$$= \left\{\mathbb{E}_{\mathbf{x}_i \sim p(\mathbf{x}_i|\mathbf{x}_o)}\left[H(p(\mathbf{z}|\mathbf{x}_i, \mathbf{x}_o))\right] - \mathbb{E}_{\mathbf{x}_i \sim p(\mathbf{x}_i|\mathbf{x}_o)}\mathbb{E}_{\mathbf{x}_\phi \sim p(\mathbf{x}_\phi|\mathbf{x}_i, \mathbf{x}_o)}\left[H(p(\mathbf{z}|\mathbf{x}_\phi, \mathbf{x}_i, \mathbf{x}_o))\right]\right\}$$
$$- \left\{\mathbb{E}_{\mathbf{x}_i \sim p(\mathbf{x}_i|\mathbf{x}_o)}\left[H(p(\mathbf{z}|\mathbf{x}_o))\right] - \mathbb{E}_{\mathbf{x}_\phi \sim p(\mathbf{x}_\phi|\mathbf{x}_o)}\left[H(p(\mathbf{z}|\mathbf{x}_\phi, \mathbf{x}_o))\right]\right\}.$$

We can show that

$$
\begin{aligned}
&H(p(\mathbf{z}|\mathbf{x}_i,\mathbf{x}_o)) - \mathbb{E}_{\mathbf{x}_\phi \sim p(\mathbf{x}_\phi|\mathbf{x}_i,\mathbf{x}_o)}\left[H(p(\mathbf{z}|\mathbf{x}_\phi,\mathbf{x}_i,\mathbf{x}_o))\right]\\
&= -\int_{\mathbf{z}} p(\mathbf{z}|\mathbf{x}_i,\mathbf{x}_o)\log p(\mathbf{z}|\mathbf{x}_i,\mathbf{x}_o)d\mathbf{z} + \int_{\mathbf{z},\mathbf{x}_\phi} p(\mathbf{z},\mathbf{x}_\phi|\mathbf{x}_i,\mathbf{x}_o)\log p(\mathbf{z}|\mathbf{x}_\phi,\mathbf{x}_i,\mathbf{x}_o)\\
&= \int_{\mathbf{z},\mathbf{x}_\phi} p(\mathbf{z},\mathbf{x}_\phi|\mathbf{x}_i,\mathbf{x}_o)\log \frac{p(\mathbf{z},\mathbf{x}_\phi|\mathbf{x}_i,\mathbf{x}_o)}{p(\mathbf{z}|\mathbf{x}_i,\mathbf{x}_o)p(\mathbf{x}_\phi|\mathbf{x}_i,\mathbf{x}_o)}\\
&= \mathscr{I}\left[\mathbf{z},\mathbf{x}_\phi|\mathbf{x}_i,\mathbf{x}_o\right],
\end{aligned}
$$

which is exactly the conditional mutual information $\mathscr{I}\left[\mathbf{z},\mathbf{x}_\phi|\mathbf{x}_i,\mathbf{x}_o\right]$ used in BALD. Therefore, our chain rule representation of reward function leads us to

$$
R(i,\mathbf{x}_o) = \mathbb{E}_{\mathbf{x}_i \sim p(\mathbf{x}_i|\mathbf{x}_o)}\mathscr{I}\left[\mathbf{z},\mathbf{x}_\phi|\mathbf{x}_i,\mathbf{x}_o\right] - \mathbb{E}_{\mathbf{x}_i \sim p(\mathbf{x}_i|\mathbf{x}_o)}\mathscr{I}\left[\mathbf{z},\mathbf{x}_\phi|\mathbf{x}_o\right].
$$

