# OpenReview forum: "EDDI: Efficient Dynamic Discovery of High-Value Information with Partial VAE"
_ICLR.cc/2019/Conference_

### Official Review · AnonReviewer2 · 2018-11-02
**A nice application model but some unclear points**

**Rating:** 6
**Confidence:** 4

**Review:**

The paper proposes Partial VAE to handle missing data and a variable-wise active learning method. The model combines Partial VAE with the acquisition function to design an intelligent information acquisition system. The paper nicely combines the missing value problem with an active learning strategy to in an acquisition pipeline and demonstrate the effectiveness on several datasets.

I have following comments/questions:

1.  Does p(x_i | z) include parameters? How do these parameters be trained?

2. Does sample from p(x_i | x_o) follow by sampling z from q(z|x_o) then sample x_i from p(x_i | z)? How to sample from p(x_\phi | x_i, x_o) in Eq (7)?

3. In Eq (9), it uses q(z_i|x_o), q(z_i | x_i, x_o),  q(z_i | x_i, x_o, x_\phi) while in Eq (4) it only shows how to learn q(z|x_o). Does it need to learn multiple partial inference networks for all combination of i and \phi ?

4. The comparison with similar algorithms seems to be weak in the experiment section. RAND is random feature selection, and SING is global feature selection by using the proposed method. These comparison methods cannot provide enough information on how well the proposed methods performs. There are plenty of works in the area of “active feature acquisition” and also many works in feature selection dated back to Lasso which should be considered as comparison targets.

5. In the “personalized” implementation of EDDI on each data instances, is the model trained independently for each data point or share some parameters across different data? If so, what are the shared parameters?

---

> ### Author Response · Authors · 2018-11-19
> **Thank you for your reviews. We have reivised the paper accoridingly and added the new baseline.**
>
> We thank you for your support of our work and valuable feedback. We have clarified all the concerns accordingly in the paper.  The original review is indented using >.
>
>
> > 1.  Does p(x_i | z) include parameters? How do these parameters be trained?
>
> Yes,  p(x_i|z) is the generator component of our partial VAE, and is trained by optimizing the partial variational bound on already observed data (with missing data), we have clarified this in our revised version of the paper .
>
> > 2. Does sample from p(x_i | x_o) follow by sampling z from q(z|x_o)
> > then sample x_i from p(x_i | z)? How to sample from p(x_\phi | x_i,
> > x_o) in Eq (7)?
>
> Yes, to sample from p(x_i | x_o), we first sample z from q(z|x_o), and then sample x_i from p(x_i | z). In the case of p(x_\phi | x_i,x_o) in Eq (7), as indicated after Eq(8), we first sample z from q(z|x_o, x_i), then sample p(x_\phi|z), since x_\phi is also an element of the set of all possible variables, x.
>
>
> > 3. In Eq (9), it uses q(z_i|x_o), q(z_i | x_i, x_o),  q(z_i | x_i,
> > x_o, x_\phi) while in Eq (4) it only shows how to learn q(z|x_o). Does
> > it need to learn multiple partial inference networks for all
> > combination of i and \phi ?
>
> No, it does not. This is one of the main novelty of our approach: we call our partial VAE approach an "amortized" partial inference method since our partial VAE parameterization of q(z|x_o) is able to handle all possible lengths of features to be conditioned on. During training (due to missingness in data), the lengths of x_o^n (where we use $n$ to indicate the index of the data point in the training set here) are different from each other. This gives q(z|x_o) the ability to generalize to  q(z_i|x_o), q(z_i | x_i, x_o),  and q(z_i | x_i,x_o, x_\phi) on test data during test time, without the need for training multiple networks.
>
> > 4. The comparison with similar algorithms seems to be weak in the
> > experiment section. RAND is random feature selection, and SING is
> > global feature selection by using the proposed method. These
> > comparison methods cannot provide enough information on how well the
> > proposed methods performs. There are plenty of works in the area of
> > “active feature acquisition” and also many works in feature selection
> > dated back to Lasso which should be considered as comparison targets.
>
> Thank you for your suggestion.
>
> We have added a new baseline adapted from LASSO in Appendix B.2.6 with UCI dataset, since LASSO requires fully observed data, and only works in problems with one-dimensional outputs. As LASSO is linear and non-probabilistic, for more fair comparison, we use the set of selected features returned by the LASSO to construct variable selection strategy and use the Partial VAE to evaluate predictive likelihoods. Please refer to our revised paper (Section 4.2) for details and results.
>
> We would like to point out that our framework is different traditional feature selection such as LASSO.  For traditional feature selection methods,  they are non-sequential and they require fully observed dataset for both training and testing which is not the case for our problem setting. Additionally, their goal is also to choose a global subset of features from fully observed data to obtain the best performance instead of select the most informative feature with any given partial observation.
>
> Our problem setting also differs from active feature acquisition (AFA) methods. As discussed in our revised paper (Section 2.2),  AFA mainly studies the optimization of *optimal training set* that would result in the best classifier (model), under limited budget of costs. On the contrary, our framework studies the problem: given a pretrained model, how to identify and acquire high value information under uncertainty, with minimal costs. Hence, AFA can not be directly applied and compared. Also, AFA requires fully observed variables at test time, while our framework does not require this assumption. Last but not least, the realization of these framework relies on various heuristics and suffer from very limited scalability.  To the best of our knowledge, DRAL is the only prior work that shares the same problem setting. We have only compared DRAL to our EDDI on a single UCI dataset since DRAL is not scalable.
>
> > 5. In the “personalized” implementation of EDDI on each data
> > instances, is the model trained independently for each data point or
> > share some parameters across different data? If so, what are the
> > shared parameters?
>
> The Partial VAE part of EDDI is trained on the training set. In the active variable selection experiments, all test data that are used to evaluate EDDI are never seen by the model before.  All model parameters are shared across different data points. In our paper, "personalized" simply means we evaluate Equation (9) on each data point individually.
>
>
> We hope that we have fully addressed your concerns in the current revised version of the paper. Please let us know if you have further questions.

---

### Official Review · AnonReviewer3 · 2018-11-05
**EDDI: EFFICIENT DYNAMIC DISCOVERY OF HIGH-VALUE INFORMATION WITH PARTIAL VAE**

**Rating:** 5
**Confidence:** 4

**Review:**

The authors present an information discovery approach based on (partial) variational autoencoders and an information theoretic acquisition function that seeks to maximize the expected information gain over a set of unobserved variables. Results are presented on image inpainting, UCI datasets and health data, namely ICU and NHANES.

It is not clear why multiple recurrent steps improve perfromance. This is not conceptually justified and empirically (see Figure 8), it is also unclear whether PNP5 significantly outperforms PNP1. Further, results seem to support that PNP is always better than PN, so why introduce the methodology around PN or even present it at all. Note that the authors do not offer an explanation about the perfromance differences between PN and PNP.

In the inpainting regions section, the authors write about well-calibrated uncertainties without any context. What do they mean by calibration, well-calibrated and how can they support their claim about it?

In Figure 3 it is not clear that PNP+Ours outperforms PNP+SING. For Boston hosing seems to be marginally better but the error bars (which I assume are standard deviations, not stated) make difficult to ascertain whether the differences are significant. Although I understand the value of having "personalized" decisions, one wonders whether this personalization comes with any generalizable measurable gains given the results.

The results in Table 2 need to be clarified and further explained. 1) what are the error bars, considering multiple runs and datasets? 2) How can EDDI be so much better than SING when individual AUICs in Tables 6-11, the only significant difference (accounting for error bars) is on Boston data? 3) according to Tables 6-11, PNP is only the best in 1 of 5 datasets, so how come is the overall beast by a large margin? This being said, the results in Table 2 are at best misleading.

In Table 4, how can PNP-EDDI be so much better than PNP-SING, when in Figure 6 error bars overlap almost everywhere?

I enjoyed reading the paper, the motivation is clear and the problem is important. The approach is modestly novel compared to existing approaches and in general well explained despite the fact that the need for multiple recurrent steps is not well justified and the differences between PN and PNP, advantages/disadvantages and when to use each are not described or explored in the experiments.

---

> ### Author Response · Authors · 2018-11-19
> **Comment Part 1: on methodologies**
>
> We thank reviewer 3 for the constructive review. We have replied the concerns to clarify possible misunderstandings and updated the paper accordingly. The original review is indented using >.
>
>
> > It is not clear why multiple recurrent steps improve perfromance. This
> > is not conceptually justified and empirically (see Figure 8), it is
> > also unclear whether PNP5 significantly outperforms PNP1. Further,
> > results seem to support that PNP is always better than PN, so why
> > introduce the methodology around PN or even present it at all.
>
> Thanks for your comment. We have agreed that the multiple recurrent steps is not crucial for performance improvement. Importantly, we agree that for the whole framework, the recurrent structure of PN is not critical for the presentation of the entire EDDI framework. Following your advice, we have replaced all PN5 result with the PN1 result in all the experiments in the paper. We have moved the recurrent PN to the Appendix as a possible extension and added a short discussion on whether one should use the recurrent version.
>
> > Note that the authors do not offer an explanation about the performance
> > differences between PN and PNP.
>
> We treat PN and PNP are two different settings of our framework. In our experiments, PNP setting performs better than PN setting for most of the evaluation.  Additionally, we have analyzed the use of PNP structure in Appendix C.1. In short, we have shown that PNP parameterization actually combines ZI-VAE (which dominates the applications of VAEs on missing data)  with PN-VAE. Therefore, we expect that PNP will enjoy the advantages from both PN and ZI, hence improve the performance. This conjecture is confirmed in the experimental results that you have mentioned.
>
>
> > In the inpainting regions section, the authors write about
> > well-calibrated uncertainties without any context. What do they mean
> > by calibration, well-calibrated and how can they support their claim
> > about it?
>
> Thanks for pointing this out. We changed to “better-estimated uncertainties” instead of “well-calibrated uncertainties” to be more technically precise in the revised version of the paper.  We have also added more explanation about it. In this case, the term "better-estimated uncertainty" is reflected by the quality of the samples generated from p(x_U|x_O). Therefore, the quality of model uncertainty is quantitatively evaluated by the test ELBO (available in Table 1, and visualized in Figure 2) of inpainting on the partially observed MNIST dataset (averaged over test set). This is calculated by $1/(N) \sum_{n=1}^{N}  ELBO(n|x_O) $, where N is the size of the test set, and ELBO(n|x_O) corresponds to the conditional ELBO of the n-th data point (where the inference net q is conditioned on x_O). Please refer to the revised paper for details.
>
> > In Figure 3 it is not clear that PNP+Ours outperforms PNP+SING. For
> > Boston hosing seems to be marginally better but the error bars (which
> > I assume are standard deviations, not stated) make difficult to
> > ascertain whether the differences are significant. Although I
> > understand the value of having "personalized" decisions, one wonders
> > whether this personalization comes with any generalizable measurable
> > gains given the results.
>
> Thank you for the comment. In all figures in the revised version of the paper,  error bars represent standard errors.  We have also performed the significance test and reported the results in Appendix B.2.3 (explained in detail for our reply for the later comments).  Our method is significantly better.
>
> Additionally,  if you also look at the enlarged subplot included in Fig 3 and Figure 9 in Appendix B.2.4, it is generally significant that the PNP+Ours curve is below the PNP+SING. The first variable selection step should be ignored when conducting such comparison since in theory, both methods should select exactly the same variable.
>
> Here we would also like to first emphasize that the proposed SING-ordering method is already a very strong alternative setting of our proposed method. First, it makes use of the same Partial VAE information of our personalized method. Secondly,  in SING-ordering, it assumes that the whole test set is available *at the same time*:  the objective of the SING is to find the average information reward for the *whole test set* at each step, which is very unrealistic in practice. This gave SING unfair advantages over EDDI.

---

> ### Author Response · Authors · 2018-11-19
> **Comment Part 2 On significance of experimental results**
>
>
> > The results in Table 2 need to be clarified and further explained. 1)
> > what are the error bars, considering multiple runs and datasets?
>
> We have revised accordingly in the paper. Regarding the error bars: in Table 2, for each run, we run all active learning strategies on each data point of each dataset. Then, we rank all strategies on an individual basis, which gives us $R * (\sum_j N_j)$ different rankings, where N_j is the size of the test set in the j-th dataset, and R is the number of runs. Finally, we simply compute the mean and standard error statistics based on these individual rankings. This procedure is explained in detail in section 4.2 in our revised version.
>
> >2) How can EDDI be so much better than SING when individual AUICs in
> > Tables 6-11, the only significant difference (accounting for error
> > bars) is on Boston data?
>
> This is a good question. We have included discussions regarding this issue in Appendix B.2.2 and B.2.3. In particular, it seems that the avg. AUIC results in Tables 6-11  contradicts the avg. ranking of AUIC results in Table 2 of the main text. However, this is not the case. In Tables 6-11,  AUIC numbers only provide a simplified statistics of *marginal performances* of each method.
>
> On the contrary, the performance comparison problem is an example of the so-called *paired samples*, which refers to the situation that different algorithms are evaluated on exactly the same set of test data points. This introduces correlations between the performances of different algorithms. The average AUIC ranking measure actually takes into account this *joint performance* of all methods, meaning that ranking is a function of the performance of all methods.  With this additional information of correlations, this gives a more accurate evaluation regarding the actual performance of different methods. Notably, in the practical scenario of active variable selection, the latter setting is more sensible and fare.
>
> The above conjecture is further validated and confirmed by applying the nonparametric statistical test on the performance results, namely the Wilcoxon signed-rank significance test on the performance samples of different methods, which are detailed in Appendix B.2.2 and B.2.3. Wilcoxon test is a very powerful statistical test which includes the information of the joint distribution in *paired samples*. In our case, the term *paired samples* refers to the situation that different algorithms are evaluated on exactly the same set of test data points, which introduces correlations between the performances of different algorithms.
>
>
>
> > 3) according to Tables 6-11, PNP is only the
> > best in 1 of 5 datasets, so how come is the overall beast by a large
> > margin? This being said, the results in Table 2 are at best
> > misleading.
>
> We believe this has been addressed in our previous reply on significance.
>
> Additionally, we would like to point out that the purpose of Tables 6-11 is to provide supplementary intuitive support that, our proposed methods, i.e., EDDI (+ PNP or PN) give the best result in 4 out of 6 datasets, compared with ZI based methods that currently dominates missing data problems in generative models. Which one to choose between PN and PNP depends on the application need.
>
> > In Table 4, how can PNP-EDDI be so much better than PNP-SING, when in
> > Figure 6 error bars overlap almost everywhere?
>
> Please refer to our previous reply regarding the PNP-SING, the *joint performance* evaluation metric, and the Wilcoxon tests.
>
> > I enjoyed reading the paper, the motivation is clear and the problem
> > is important. The approach is modestly novel compared to existing
> > approaches and in general well explained despite the fact that the
> > need for multiple recurrent steps is not well justified and the
> > differences between PN and PNP, advantages/disadvantages and when to
> > use each are not described or explored in the experiments.
>
> We are grateful that you enjoyed reading the paper and point out the part of the paper that needs clarification. We hope that we have fully addressed your concerns in the current revised version of the paper. Please let us know if you have further questions.

---

### Official Review · AnonReviewer1 · 2018-11-07
**Interesting but difficult to read**

**Rating:** 6
**Confidence:** 2

**Review:**

----I acknowledge that the authors have made improvements to the paper and have increased my score to 6

This is still definitely not my area of expertise and so I am leaving my confidence score low.
---

The paper presents an algorithm EDDI that uses a a partial VAE and does active feature selection. The authors show quite a bit of experiments that seem to indicate the approach gives positive results.  However, since this is not my main area of expertise I do not know if these tasks are standard evaluation for this task.

For instance in Section 4.3, 4.4 why don't the authors plot accuracy as a function of steps/number of variables observed. That would seem much more useful than log likelihood.

In general, I found the methodology in the paper to be difficult to understand and not enough background was given.
I think the paper would be clearer if it was more self contained.

-For instance, I found much of Section 3 to not have enough background. The authors use lots of terminology around VAEs but don't give enough rigorous background so the paper doesn't feel self contained.

-The same is true regarding "amortized inference" which I also feel isn't rigorously defined anywhere but often discussed.

-The task for Section 4.1 (image inpainting) is not quite defined.

---

> ### Author Response · Authors · 2018-11-19
> **Revision uploaded**
>
> Thank reviewer 1 for appreciating and application and the positive results. We have replied the concerns to clarify possible misunderstandings and updated the paper accordingly. The original review is indented using >.
>
> > Review: The paper presents an algorithm EDDI that uses a a partial VAE and does active
> > feature selection. The authors show quite a bit of experiments that seem to indicate the
> > approach gives positive results.  However, since this is not my main area of expertise I do
> > not know if these tasks are standard evaluation for this task.
>
> > For instance in Section 4.3, 4.4 why don't the authors plot accuracy as a function of
> > steps/number of variables observed. That would seem much more useful than log
> > likelihood.
>
> Apart from existing results, we have reported test RMSE as suggested in the Appendix.B.2.5 for all UCI experiments in the revised version of the paper. Accuracy in terms of RMSE is consistent with the reported result using predictive log likelihood. Additionally, we would like to clarify that, log likelihood is the common standard when evaluating the performance related to generative models [1,2]. Compared with accuracy metric such as RMSE, log likelihood also account for model uncertainties (of the posterior on latent variable, z), which is very crucial in the practical application of active variable learning.
>
> Reference:
> [1] Kingma, Diederik P., and Max Welling. "Auto-encoding variational Bayes." arXiv preprint arXiv:1312.6114 (2013).
> [2] Gregor, Karol, et al. "Draw: A recurrent neural network for image generation." arXiv preprint arXiv:1502.04623 (2015).
> [3] Kingma, Diederik P., and Prafulla Dhariwal. "Glow: Generative flow with invertible 1x1 convolutions." arXiv preprint arXiv:1807.03039 (2018).
>
> > In general, I found the methodology in the paper to be difficult to understand and not enough background was given.
> > I think the paper would be clearer if it was more self-contained.
>
> > For instance, I found much of Section 3 to not have enough background. The authors use
> > lots of terminology around VAEs but don't give enough rigorous background so the paper
> > doesn't feel self contained.
>
> > The same is true regarding "amortized inference" which I also feel isn't rigorously defined
> > anywhere but often discussed.
>
> Thanks for your comment. We have revised the paper and added a paragraph “VAE and amortized inference” in section 3.2 which is a brief, self-contained introduction to VAEs and amortized inference.
>
> > The task for Section 4.1 (image inpainting) is not quite defined.
>
> We have added a short description in 4.1 to make the task more clarified and well-defined.

---

### Author Response · Authors · 2018-11-19
**Summary of the new revision**

Dear all,

We have revised our paper utilizing all the feedback. We thus summarize main the changes in our revised paper below for your references.

* Based on the comment of Reviewer 3, we have moved the introduction and discussion of recurrent PNs to the Appendix B.5. We have also revised the presentation of the Partial VAE include Figure 1. We have also updated all the experimental results with non-recurrent PN in the main paper.
* We have added statistical tests for model comparison in Appendix B. 2. (Reviewer 3), all the improvement is statistically significant.
* We have added RMSE plots in Appendix B.2.5, as requested by Reviewer 1 and show that using RMSE, the conclusion is consistent as using predictive likelihood.
* We have discussed a new baseline that utilizing lasso in Appendix B.2.5, as suggested by Reviewer 2;
* We have added a discussion on the active feature acquisition (AFA) in Section 2.2 and clarified the different of AFA and our problem setting.
* We have added the introduction of VAEs and amortized inference in Section 3.2, as required by Reviewer 1;
* We have added brief descriptions of image inpainting task in Section 4.1 (Reviewer 1).

---

### Meta-Review · Area_Chair1 · 2018-12-18
**valuable baselines though lots of room for improvement**

**Confidence:** 3
**Recommendation:** Reject

**Metareview:**

This paper develops an active variable selection framework that couples a partial variational autoencoder capable of handling missing data with an information acquisition criteria derived from Bayesian experimental design. The paper is generally well written and the formulation appears to be natural, with a compelling real world healthcare application. The topic is relatively under-explored in deep learning  and the paper appears to attempt to set a valuable baseline. However, the AC cannot recommend acceptance based on the fact that reviewer 2 has brought up concerns about the competitiveness of the approach relative to alternative methods reported in the experimental section, and all reviewers have found various parts of the paper to have room for improvement with regards to technical clarity. As such the paper would benefit from a revision and a stronger resubmission.